# Improved $\ell_p$ Regression via Iteratively Reweighted Least Squares

**Alina Ene**
Department of Computer Science
Boston University
aene@bu.edu

**Ta Duy Nguyen**
Department of Computer Science
Boston University
taduy@bu.edu

**Adrian Vladu**
CNRS & IRIF
Université Paris Cité
vladu@irif.fr

## Abstract

We introduce fast algorithms for solving $\ell_p$ regression problems using the iteratively reweighted least squares (IRLS) method. Our approach achieves state-of-the-art iteration complexity, outperforming the IRLS algorithm by Adil-Peng-Sachdeva (NeurIPS 2019) and matching the theoretical bounds established by the complex algorithm of Adil-Kyng-Peng-Sachdeva (SODA 2019, J. ACM 2024) via a simpler lightweight iterative scheme. This bridges the existing gap between theoretical and practical algorithms for $\ell_p$ regression. Our algorithms depart from prior approaches, using a primal-dual framework, in which the update rule can be naturally derived from an invariant maintained for the dual objective. Empirically, we show that our algorithms significantly outperform both the IRLS algorithm by Adil-Peng-Sachdeva and MATLAB/CVX implementations.

## 1 Introduction

In this paper, we study the $\ell_p$ regression problem defined as follows. The input to the problem is a matrix $A \in \mathbb{R}^{d \times n}$, a vector $b \in \mathbb{R}^d$ that lies in the column span of $A$, and an accuracy parameter $\epsilon$. The goal is to approximately solve the problem $\min_{x \in \mathbb{R}^n : Ax = b} \|x\|_p$, i.e., find a solution $x \in \mathbb{R}^n$ such that $Ax = b$ and $\|x\|_p \leq (1 + \epsilon)\|x^*\|_p$, where $x^*$ is an optimal solution to the problem, and $\|\cdot\|_p$ denotes the $\ell_p$ norm. Solving $\ell_p$ regression for all values of $p$ is a fundamental problem in machine learning with numerous applications and has been studied in a long line of research beyond the classical least squares regression with $p = 2$. $L_p$-norm regression problems with general $p$ arise in several areas, including supervised learning, graph clustering, and wireless networks. Examples of applications include $\ell_p$-norm based algorithms in semi-supervised learning (Alaoui, 2016; Liu and Gleich, 2020), $k$-clustering with $\ell_p$-norm (Huang and Vishnoi, 2020), robust regression and robust clustering (Meng and Mahoney, 2013; Huang et al., 2023).

For this general class of convex optimization problems, designing provably fast iterative algorithms to obtain high accuracy solutions with empirical efficiency is an important question. General convex programming methods such as interior point methods are usually slow in practice. In theory, Bubeck et al. (2018) show that algorithms based on interior point methods cannot improve beyond $O(\sqrt{n})$ iterations[1] for any $p \notin \{1, 2, \infty\}$. Breaking this barrier and finding iterative algorithms that are faster than interior point methods both in theory and practice is the goal of this line of work.

Recent developments have led to new algorithmic approaches such as a homotopy method (Bubeck et al., 2018), and an iterative refinement approach (Adil et al., 2019a;b; 2024) for $\ell_p$ regression with $p \notin \{1, \infty\}$. We highlight the notable works by Adil et al. (2019a;b; 2024). On the one hand, the algorithm with the best known theoretical runtime is given by Adil et al. (2019a; 2024) with

---

[1]For simplicity in the introduction, we assume that $d = \Theta(n)$. In the regime when $n \gg d$, the IPM iteration complexity improves to $\widetilde{O}(\sqrt{d})$.

$O\big(p^2 n^{\frac{p-2}{3p-2}} \log\big(\frac{n}{\epsilon}\big)\big)$ calls[2] to a linear system solver. This algorithm, however, relies on complex subroutines and includes theoretical choices for several hyperparameters. In practice, to obtain an efficient implementation, hyperparameters require tuning. Due to these reasons, this theoretical algorithm by Adil et al. (2019a; 2024) does not provide a practical implementation. On the other hand, an algorithm known as $p$-IRLS by Adil et al. (2019b) has been shown to have significant speed up over standard solvers such as CVX. This algorithm is implemented based on an Iteratively Reweighted Least Squares (IRLS) method, which is a general iterative framework for solving regression problems. The key element of an IRLS method is solving a weighted least squares regression problem in each iteration. This is equivalent to solving a linear system of the form $\min_{x\in\mathbb{R}^n:Ax=b} x^\top R x$, where $R$ is a diagonal matrix, which can be computed very efficiently in practice with the advance of numerical solvers. IRLS algorithms are favored in practice (Burrus, 2012), but designing IRLS algorithms with strong convergence guarantees is challenging. In particular, to obtain the efficiency, the algorithm by Adil et al. (2019b) sacrifices the theoretical guarantee, requiring $O\big(p^3 n^{\frac{p-2}{2p-2}} \log\big(\frac{n}{\epsilon}\big)\big)$ linear system solves. This brings forth the question:

*Can we design an algorithm that retains the empirical efficiency of an IRLS approach while achieving the state-of-the-art theoretical runtime?*

In this work, we give a positive answer to this question. We provide a new algorithmic framework for $\ell_p$ regression based on an IRLS approach for all values of $p \in (1,\infty)$. We propose an algorithm that uses $O\big(p^2 n^{\frac{p-2}{3p-2}} \log\big(\frac{n}{\epsilon}\big)\big)$ linear system solves, matching the state-of-the-art theoretical algorithm by Adil et al. (2019a), and improving upon the guarantee of $O\big(p^3 n^{\frac{p-2}{2p-2}} \log\big(\frac{n}{\epsilon}\big)\big)$ for the $p$-IRLS algorithm by Adil et al. (2019b). We experimentally compare our algorithm with the $p$-IRLS algorithm (Adil et al., 2019b) and CVX solvers, and we observe significant improvements in all instances.

## 1.1 OUR CONTRIBUTIONS

For the simplicity of the exposition, we study the $\ell_p$ regression problem in both low and high precision regimes for $p \geq 2$.

*Remark* 1.1. In Appendix B, we show a simple reduction for the more general problem $\min_{x:\ Ax=b} \|Nx - v\|_p$ to the form $\min_{x:\ \tilde{A}x=\tilde{b}} \|x\|_p$ with the dependence of the runtime on the number of rows of $N$ instead of the dimension of $x$. We also show in Appendix C a reduction for the case $1 < p < 2$ to the case $p \geq 2$.

In the low precision regime when the runtime dependence on $\epsilon$ is $\mathrm{poly}\big(\frac{1}{\epsilon}\big)$, we have the following theorem.

**Theorem 1.1.** *For any $p \geq 2$, there is an iterative algorithm for the $\ell_p$ regression problem $\min_{x\in\mathbb{R}^n:\ Ax=b} \|x\|_p$ that solves $O\left(\log\log n + \log(1/\epsilon)\right)$ subproblems, each of which makes $O\left(\left(\big(\frac{1}{\epsilon}\big)^{\frac{2p-3}{p-2}} + n^{\frac{p-2}{3p-2}}\big(\frac{1}{\epsilon}\big)^{\frac{3p^2-4p}{3p^2-8p+4}}\right)\log\big(\frac{n}{\epsilon^{\frac{p}{p-2}}}\big)\right)$ calls to solve a linear system of the form $ADA^\top \phi = b$, where $D$ is an arbitrary non-negative diagonal matrix.*

*Remark* 1.2. When $p = \infty$, each subproblem makes $O\big(\frac{1}{\epsilon^2} + \frac{n^{\frac{1}{3}}}{\epsilon} \log(\frac{n}{\epsilon})\big)$ calls to a linear system solver.

Prior approaches for solving $\ell_p$ regression problem in the low precision regime commonly use the Taylor expansion of $\|x\|_p^p$, which then allows for deriving and bounding the updates. In contrast to this, our algorithm relies on a primal-dual approach using the dual formulation of the squared objective $\min_{x:\ Ax=b} \|x\|_p^2 = \min_{x:\ Ax=b} \|x^2\|_{p/2} = \max_r \frac{\mathcal{E}(r)}{\|r\|_q}$ where $\ell_q$ is the dual norm of $\ell_{p/2}$ and $\mathcal{E}(r) = \min_{x:Ax=b}\langle r, x^2\rangle$. The term $\mathcal{E}(r)$ is often referred to as the energy. The high level idea of our approach is as follows. Starting with an initial solution $r$ for the dual problem, we will increase the coordinates of $r$ as much as possible so that the increase in the energy $\mathcal{E}(r)$ relative to the increase

---

[2]The original result is $O\big(pn^{\frac{p-2}{3p-2}} \log\big(\frac{\|x^{(0)}\|_p^p - \|x^*\|_p^p}{\epsilon}\big)\big)$ for finding $\widehat{x}$ such that $\|\widehat{x}\|_p^p \leq \min_{x:Ax=b} \|x\|_p^p + \epsilon$. This translates to $O\big(pn^{\frac{p-2}{3p-2}} \log\big(\frac{\|x^{(0)}\|_p^p - \|x^*\|_p^p}{p\epsilon\|x^*\|_p^p}\big)\big) = O\big(p^2 n^{\frac{p-2}{3p-2}} \log\big(\frac{n}{\epsilon}\big)\big)$ for finding $\widehat{x}$ such that $\|\widehat{x}\|_p \leq (1+\epsilon)\min_{x:Ax=b} \|x\|_p$ for $x^{(0)}$ initialized to $\min_{x:Ax=b} \|x\|_2$.

of $\|r\|_q$ is also sufficiently large, until we can obtain a $(1 - \epsilon)$ optimal dual solution and whereby recover an approximately optimal primal solution. This template is close to the approach for $\ell_\infty$ regression by Ene and Vladu (2019). However, $\ell_p$ regression does not have the readily decomposable structure along the coordinates as $\ell_\infty$ regression and novel technique is required in the design of the algorithm. Our approach is also a reminiscence of the width-independent multiplicative weights update method for solving mixed packing covering linear program, where in each step the algorithm updates the coordinates the maximize the bang-for-buck ratio (Quanrud, 2020). In contrast to MWU, we do not use a mirror map or regularize $\ell_p$ norms to make them smooth as in standard approaches. Our scheme allows our method to take much longer steps, where in each step, the coordinates of the dual solution are allowed to change by large polynomial factors and thereby achieve faster running time.

To obtain faster algorithms in the high accuracy regime with a logarithmic dependence on the accuracy, we adapt the iterative refinement approach of Adil et al. (2019a) and obtain improved running times.

**Theorem 1.2.** *For any $p \geq 2$, there is an iterative algorithm for the $\ell_p$ regression problem $\min_{x \in \mathbb{R}^n : Ax = b} \|x\|_p$ that solves $O\left(p^2 \log n \log\left(\frac{n}{\epsilon}\right)\right)$ subproblems, each of which makes $O\left(n^{\frac{p-2}{3p-2}}\right)$ calls to solve a linear system of the form $\widetilde{A} D \widetilde{A}^\top \phi = z$, where $D$ is an arbitrary non-negative diagonal matrix, $\widetilde{A}$ is a matrix obtained from $A$ by appending a single row, and $z$ is a vector obtained from the all-zero vector by appending a single non-zero coordinate.*

Using the iterative refinement template by (Adil et al., 2019a;b; 2024), we instead use an IRLS solver for the residual problems with improved runtime. The residual solver solves a mixed $\ell_p + \ell_2$ problem in the form $\min_{x : Ax = b} \|x\|_p^2 + \langle \theta, x^2 \rangle$, only to a constant approximation. Here the challenge lies in the fact that the $\ell_2$ term makes the dual problem no longer scale-free and thus our low precision solver is not immediately usable. However, by an appropriate initialization of the dual solution and careful adjustments to the step size, our algorithm achieves the desired $O\left(n^{\frac{p-2}{3p-2}}\right)$ bound. Since regularized $\ell_p + \ell_2$ regression problems arise in many applications in machine learning and beyond, our algorithm for the mixed $\ell_p + \ell_2$ objective is of independent interest.

Finally, we experimentally evaluate our high-precision algorithm. Our algorithm significantly outperforms the $p$-IRLS algorithm (Adil et al., 2019a) both in the number of linear system solves as well as the overall running time. Our algorithm is significantly faster than CVX solvers and is able to run on large instances, which is not possible for CVX solvers within a time constraint.

## 1.2 RELATED WORK

$\ell_p$ regression problems have received significant attention. Here we summarize the results that are closest to our work. The surveyed algorithms are iterative algorithms where the running time of each iteration is dominated by a single linear system solve.

Algorithms based on interior point methods use $\widetilde{O}\left(\sqrt{n}\right)$ iterations for any $p \in [1, \infty]$ (Nesterov and Nemirovskii, 1994), which was improved to $\widetilde{O}\left(\sqrt{d}\right)$ iterations for $p \in \{1, \infty\}$ (Lee and Sidford, 2014). Bubeck-Cohen-Lee-Li (Bubeck et al., 2018) show that this iteration bound is generally necessary for interior point methods and propose a homotopy-based algorithm that uses $\tilde{O}\left(\text{poly}\left(\frac{p^2}{p-1}\right) \cdot n^{|1/2 - 1/p|}\right)$ iterations for any $p \notin \{1, \infty\}$. Adil et al. (2019a; 2024) introduced an iterative refinement framework that uses $O\left(p^2 \cdot n^{\frac{p-2}{3p-2}} \log\left(\frac{n}{\epsilon}\right)\right)$ iterations for any $p > 2$. Using Lewis weight sampling, Jambulapati-Liu-Sidford (Jambulapati et al., 2022) improve the method by Adil et al. (2019a; 2024) to $O\left(p^p \cdot d^{\frac{p-2}{3p-2}} \text{polylog}\left(\frac{n}{\epsilon}\right)\right)$, for overconstrained regression problems $\min_{x \in \mathbb{R}^d} \|Ax - b\|_p$ where $A \in \mathbb{R}^{n \times d}$ and $n$ is much larger than $d$ (the iteration complexity of the prior algorithms will still depend on the larger dimension $n$ in this case). Bullins (2018) gives a faster algorithm for minimizing structured convex quartics, which implies an algorithm for $\ell_4$ regression with $\tilde{O}(n^{\frac{1}{5}})$ iterations. Building on the work of Christiano et al. (2011); Chin et al. (2013) for maximum flows and regression, Ene and Vladu (2019) give an algorithm for $\ell_1$ and $\ell_\infty$ regression using $O\left(\frac{n^{1/3} \log(1/\epsilon)}{\epsilon^{2/3}} + \frac{\log n}{\epsilon^2}\right)$ iterations. This work also uses a primal-dual framework but the algorithm and analysis are specific to the special structure of the $\ell_1$ and $\ell_\infty$ norm and work only in the low precision regime with $\text{poly}\left(\frac{1}{\epsilon}\right)$ convergence.

---

**Algorithm 1** $\ell_{2p}$-minimization$(A, b, \epsilon)$

---

**Input:** Matrix $A \in \mathbb{R}^{d \times n}$, vector $b \in \mathbb{R}^d$, accuracy $\epsilon$
**Output**: Vector $x$ such that $Ax = b$ and $\|x\|_{2p} \leq (1 + \epsilon) \min_{x:Ax=b} \|x\|_{2p}$
Initialize $x^{(0)} = \min_{x:Ax=b} \|x\|_2$
$L = \max \left\{ i : (1 + \epsilon)^i \leq \frac{\|x^{(0)}\|_2}{n^{\frac{1}{2} - \frac{1}{2p}}} \right\}; U = \min \left\{ i : (1 + \epsilon)^i \geq \|x^{(0)}\|_2 \right\}$
**while** $L < U$:
    $P = \lfloor \frac{L+U}{2} \rfloor$, $M = (1 + \epsilon)^P$
    **if** $\mathrm{SubSolver}(A, b, \epsilon, M)$ is infeasible **then**
        $L = P + 1$
    **else**
        Let $x^{(t+1)}$ be the output of $\mathrm{SubSolver}(A, b, \epsilon, M)$
        $U = P; t \leftarrow t + 1$
    **end if**
**end while**
**return** $x^{(t)}$

---

**Algorithm 2** $\mathrm{SubSolver}(A, b, \epsilon, M)$

---

**Input:** Matrix $A \in \mathbb{R}^{d \times n}$, vector $b \in \mathbb{R}^d$, accuracy $\epsilon$, target value $M$
**Output**: Vector $x$ such that $Ax = b$ and $\|x\|_{2p} \leq (1 + \epsilon)M$,
        or approximate infeasibility certificate $r$, $\|r\|_q = 1$.
$t = 0, r^{(0)} = \frac{1}{n^{1/q}}, t' = 0, s^{(t')} = 0$
**while** $\left\| \left( r^{(t)} \right) \right\|_q \leq \frac{1}{\epsilon}$
    $x^{(t)} = \arg\min_{x:Ax=b} \langle r^{(t)}, x^2 \rangle$
    $\gamma_i^{(t)} = \begin{cases} \frac{x_i^2 \|r\|_q^{q-1}}{M^2 r_i^{q-1}} & \text{if } \frac{x_i^2 \|r\|_q^{q-1}}{r_i^{q-1}} \geq (1 + \epsilon)M^2 \\ 1 & \text{otherwise} \end{cases}$, for all $i$
    **if** $\gamma^{(t)} = 1$ **then return** $x^{(t)}$ **end if**                                                  ▷ *Case 1*
    $\alpha^{(t)} = \left( \gamma^{(t)} \right)^{\frac{1}{q}}; r^{(t+1)} = r^{(t)} \cdot \alpha^{(t)}$
    **if** $\alpha^{(t)} \leq n^{\frac{2}{2q+1}} \left( \frac{1}{\epsilon} \right)^{\frac{q-1}{2q+1}}$ **then** $s^{(t'+1)} = s^{(t')} + x^{(t)}; t' = t' + 1$ **end if**
    **if** $t' > 0$ **and** $\left\| s^{(t')}/t' \right\|_{2p} \leq (1 + \epsilon)M$ **then return** $s^{(t')}/t'$ **end if**    ▷ *Case 2*
    $t = t + 1$
**end while**
**return** $r^{(t)}$                                                                                            ▷ *Case 3*

---

## 2  OUR ALGORITHM WITH poly $\left( \frac{1}{\epsilon} \right)$ CONVERGENCE

In this section, we present our algorithm with guarantee provided in Theorem 1.1.

Before describing the algorithm, we first introduce some basic notations. For a constant $a \in \mathbb{R}$, we abuse the notation and use $a \in \mathbb{R}^n$ to denote the vector with all entries equal to $a$ (the dimension will be clear from context). When it is clear from the context, we apply scalar operations to vectors with the interpretation that they are applied coordinate-wise. For $p \geq 1$, we let $q$ be such that $\frac{1}{p} + \frac{1}{q} = 1$ and $\ell_q$ is the dual norm of the $\ell_p$ norm.

### 2.1  OUR ALGORITHM

For ease of notation, it is convenient to consider the following equivalent formulation of the problem: For $p \geq 1$, we solve $\min_{x:Ax=b} \|x\|_{2p}^2 = \min_{x:Ax=b} \|x^2\|_p$ to $(1 + \epsilon)$ multiplicative error. We provide our algorithm in Algorithms 1 and 2. We give an overview of our approach and explain the intuition in the following section.

## 2.2 OVERVIEW OF OUR APPROACH

Our algorithm is based on a primal-dual approach, starting with the following dual formulation of the problem. Using $q$ as the dual norm of $p$ and by duality, we write

$$\min_{x:Ax=b} \|x\|_{2p} = \min_{x:Ax=b} \left\|x^2\right\|_p = \min_{x:Ax=b} \max_{r:\|r\|_q \leq 1} \langle r, x^2 \rangle \max_{r \geq 0:\|r\|_q \leq 1} \min_{x:Ax=b} \langle r, x^2 \rangle = \max_{r \geq 0} \frac{\mathcal{E}(r)}{\|r\|_q},$$

where we defined $\mathcal{E}(r) := \min_{x:Ax=b} \langle r, x^2 \rangle$. The main part of our algorithm is the subroutine shown in Algorithm 2, which takes as input a guess $M$ for the optimum value $\|x^*\|_{2p}$. To find an $(1 + \epsilon)$ approximation of the optimum value, the main Algorithm 1 performs a binary search as follows. Since $x^{(0)}$ is initialized to $\min_{x:Ax=b} \|x\|_2$, we can show that $\|x^*\|_p$ is contained in the range $\left[ \frac{\|x^{(0)}\|_2}{n^{\frac{1}{2} - \frac{1}{2p}}}, \|x^{(0)}\|_2 \right]$. The algorithm performs binary search over the indices $i$ such that $(1 + \epsilon)^i$ is in that range. Note that the main algorithm only needs to perform at most $\log \left( \frac{\log n}{\epsilon} \right)$ iterations, each of which makes one call to the subproblem solver.

We now focus on the subproblem when we are given a guess $M$ and a target precision $\epsilon$. The goal is to find a primal solution $x$ that satisfies $\|x\|_{2p} \leq M(1 + \epsilon)$ or a dual solution $r$ (infeasibility certificate) which can certify that $\min_{x:Ax=b} \|x\|_{2p}^2 \geq \frac{\mathcal{E}(r)}{\|r\|_q} \geq (\frac{M}{1+\epsilon})^2$. This lower bound on the optimal value of the problem tells us that we can increase the guess $M$.

The objective function $\mathcal{E}(r)$ has a very useful monotonicity property: it increases when $r$ increases. The overall strategy of our algorithm is to start with an initial dual solution $r^{(0)}$ (which we initialize uniformly to $\frac{1}{n^{1/q}}$) and increase it while maintaining the following invariant

$$\mathcal{E}(r^{(t+1)}) - \mathcal{E}(r^{(t)}) \geq M^2 \left( \left\|r^{(t+1)}\right\|_q - \left\|r^{(t)}\right\|_q \right), \tag{1}$$

or equivalently,

$$\frac{\mathcal{E}(r^{(t+1)}) - \mathcal{E}(r^{(t)})}{\left\|r^{(t+1)}\right\|_q - \left\|r^{(t)}\right\|_q} \geq M^2.$$

The telescoping property of both sides of (1) will guarantee that, if the algorithm outputs a dual solution $r$ with sufficiently large $\|r\|_q$, this solution will satisfy $\mathcal{E}(r) \geq \left( \frac{M}{1+\epsilon} \right)^2 \|r\|_q$, i.e, $\frac{\mathcal{E}(r)}{\|r\|_q} \geq \left( \frac{M}{1+\epsilon} \right)^2$. To maintain the invariant 1, we have two useful bounds for the change in the objective and dual solution:

$$\mathcal{E}(r^{(t+1)}) - \mathcal{E}(r^{(t)}) \geq \sum_i r_i^{(t)} \left( x_i^{(t)} \right)^2 \left( 1 - \frac{r_i^{(t)}}{r_i^{(t+1)}} \right), \tag{2}$$

$$\frac{1}{\left\|r^{(t+1)}\right\|_q - \left\|r^{(t)}\right\|_q} \geq \frac{q \left\|r^{(t)}\right\|_q^{q-1}}{\sum_i \left( r_i^{(t+1)} \right)^q - \left( r_i^{(t)} \right)^q}. \tag{3}$$

Both inequalities allow us to decompose the invariant along the coordinates. That is, we can maintain the invariant by ensuring for each coordinate $i$ that we increase that

$$\frac{q \left\|r^{(t)}\right\|_q^{q-1} r_i^{(t)} \left( x_i^{(t)} \right)^2}{\left( r_i^{(t+1)} \right)^q - \left( r_i^{(t)} \right)^q} \left( 1 - \frac{r_i^{(t)}}{r_i^{(t+1)}} \right) \geq M^2.$$

In order to do this, we update each $r_i^{(t)}$ multiplicatively, via the term $\gamma_i^{(t)} = \frac{\left\|r^{(t)}\right\|_q^{q-1}}{\left( r_i^{(t)} \right)^{q-1}} \cdot \frac{\left( x_i^{(t)} \right)^2}{M^2}$.

To guarantee fast convergence, we want to increase $r_i^{(t)}$ as much as possible, by setting a target threshold on $\gamma_i^{(t)}$: if $\gamma_i^{(t)}$ exceeds the threshold, we update $r_i^{(t+1)} = r_i^{(t)} \left( \gamma_i^{(t)} \right)^{1/q}$; otherwise, $r_i^{(t)}$

remains unchanged. When we can no longer increase $r$ while preserving the invariant, we can be sure that we have found the corresponding primal solution $x$ with small norm. During the course of the algorithm, we also keep track of iterations with small increases in $r$ and use the uniform average over the corresponding primal solutions to obtain an approximately feasible primal solution, in case the algorithm fails to return an infeasibility certificate quickly enough.

We note that our update approach is derived in a completely different way from standard iterative frameworks such as multiplicatives weights updates and, generally, mirror descent. In contrast to these standard approaches, we do not use a mirror map or regularize $\ell_p$ norms to make them smooth. Our update scheme allows our algorithm to take much longer steps, and the coordinates of the dual solution are allowed to change by large polynomial factors in each step. This allows us to obtain a fast convergence rate.

We outline the necessary lemmas needed to prove Theorem 1.1 before providing complete analysis and proof in Appendix D.

**Correctness of Algorithm 2.** There are two possible outcomes of Algorithm 2. Either it returns a primal solution (Case 1 and Case 2) or a dual certificate (Case 3). In the former two cases, Case 2 immediately gives us an approximate solution. We show in Lemma 2.2 that the returned vector in Case 1 achieves the target approximation guarantee. In Case 3, we use the invariant shown in Lemma 2.1 to show that the returned dual solution is an infeasibility certificate.

We formalize these statements in the lemmas below.

**Lemma 2.1** (Invariant). *For all $t$, we have that if $\gamma^{(t)} \neq 1$ then $\frac{\mathcal{E}(r^{(t+1)}) - \mathcal{E}(r^{(t)})}{\left\|r^{(t+1)}\right\|_q - \left\|r^{(t)}\right\|_q} \geq M^2.$*

**Lemma 2.2** (Case 1). *Let $r$ be a dual solution and $x = \arg\min_{\widehat{x} : A\widehat{x} = b} \langle r, \widehat{x}^2 \rangle$. If $\left\| \|r\|_q^{q-1} \cdot \frac{x^2}{r^{q-1}} \right\|_\infty \leq (1 + \epsilon) M^2$ then $\|x\|_{2p} \leq M(1 + \epsilon).$*

**Lemma 2.3** (Case 3). *If the algorithm returns $r^{(T)}$, then $\frac{\mathcal{E}(r^{(T)})}{\left\|r^{(T)}\right\|_q} \geq \frac{M^2}{(1+\epsilon)^2}.$*

**Convergence of Algorithm 2.** We run the algorithm for $T$ iterations. The algorithm terminates if at any point it finds a solution $x$ that satisfies the desired bound (otherwise it is unable to further increase the dual solution). Otherwise, we show that it must finish very fast. Suppose we run it for $T = T_{hi} + T_{lo}$ iterations. Let the iterations in $T_{hi}$ correspond to those where at least a single coordinate of $r$ was scaled by $\geq S := n^{\frac{2}{2q+1}} \left(\frac{1}{\epsilon}\right)^{\frac{q-1}{2q+1}}$. Let $T_{lo}$ be the remaining iterations. The following lemmas give an upperbound on $T_{hi}$ and $T_{lo}$.

**Lemma 2.4.** *We have $T_{hi} \leq \frac{n}{S^q \epsilon^q}.$*

**Lemma 2.5.** *We have $T_{lo} \leq O\left( \left(\frac{1}{\epsilon} + \frac{S^{1/2}}{q \ln S}\right) \frac{1}{\epsilon^{\frac{q+1}{2}}} \log\left(\frac{n}{\epsilon^q}\right) \right).$*

Since $S = n^{\frac{2}{2q+1}} \left(\frac{1}{\epsilon}\right)^{\frac{q-1}{2q+1}}$, we obtain the following convergence guarantee:

**Lemma 2.6.** *Algorithm 2 terminates in $O\left( \left( \left(\frac{1}{\epsilon}\right)^{\frac{q+3}{2}} + n^{\frac{1}{2q+1}} \left(\frac{1}{\epsilon}\right)^{\frac{q^2+2q}{2q+1}} \right) \log\left(\frac{n}{\epsilon^q}\right) \right)$ iterations.*

Equipped with these lemmas, we give the proof for Theorem 1.1.

*Proof of Theorem 1.1.* Returning to the problem $\min_{x \in \mathbb{R}^n : Ax = b} \|x\|_p$, we have the main algorithm executes a binary search over the power of $(1 + \epsilon)$ in the range $\left[ \frac{\|x^{(0)}\|_2}{n^{\frac{1}{2} - \frac{1}{p}}}, \|x^{(0)}\|_2 \right]$, so the total number of calls to the subroutine solver is $O\left(\log \log n + \log \frac{1}{\epsilon}\right)$. By Lemma 2.6, the subroutine solver requires $O\left( \left( \left(\frac{1}{\epsilon}\right)^{\frac{q+3}{2}} + n^{\frac{1}{2q+1}} \left(\frac{1}{\epsilon}\right)^{\frac{q^2+2q}{2q+1}} \right) \log\left(\frac{n}{\epsilon^q}\right) \right)$ linear system solves, where $q = \frac{p}{p-2}$ is the dual norm of $p/2$. Substituting the value of $q$, we obtain the conclusion. $\square$

---

**Algorithm 3** Iteratively Reweighted Least Squares

---

**Input:** Matrix $A \in \mathbb{R}^{d \times n}$, vector $b \in \mathbb{R}^d$, $\epsilon$

**Output**: Vector $x$ such that $Ax = b$ that minimizes $\|x\|_p^p$

Initialize $x^{(0)} = \arg\min_{x:Ax=b} \|x\|_2^2$

$M^{(0)} := \frac{\|x^{(0)}\|_p^p}{16p}, t \leftarrow 0; \kappa = \begin{cases} 1 & \text{if } p \leq \frac{2\log n}{\log n - 1} \\ \frac{p}{p-2} & \text{otherwise} \end{cases}$

**while** $M^{(t)} \geq \frac{\epsilon}{16p(1+\epsilon)} \|x^{(t)}\|_p^p$

    $g^{(t)} = |x^{(t)}|^{p-2} x^{(t)}; R^{(t)} = 2 |x^{(t)}|^{p-2}$

    $\tilde{\Delta} \leftarrow \text{ResidualSolver}\left(\frac{p}{2}, \begin{bmatrix} A \\ (g^{(t)})^\top \end{bmatrix}, \left[0, \frac{M^{(t)}}{2}\right], (M^{(t)})^{\frac{2-p}{p}} R^{(t)}, 2\sqrt{\kappa}(M^{(t)})^{\frac{1}{p}}\right)$

    **if** $\tilde{\Delta}$ is an infeasibility certificate or $\left\langle R^{(t)}, \tilde{\Delta}^2 \right\rangle \geq 2M^{(t)}$ **then**

        $M^{(t+1)} \leftarrow M^{(t)}/2, x^{(t+1)} = x^{(t)}$

    **else**

        $M^{(t+1)} \leftarrow M^{(t)}, x^{(t+1)} = x^{(t)} - \frac{\tilde{\Delta}}{64p\kappa}$

    **end if**

    $t \leftarrow t + 1$

**end while**

**return** $x^{(t)}$

---

# 3 OUR ALGORITHM WITH $\log\left(\frac{1}{\epsilon}\right)$ CONVERGENCE

## 3.1 ALGORITHM

In this section, we present our algorithm with guarantee provided in Theorem 1.2. For the ease of the exposition, we consider a slight variation of the problem: for $p \geq 2$, we solve $\min_{x:Ax=b} \|x\|_p^p$ to $(1 + \epsilon)$ multiplicative error. We show our algorithm in Algorithms 3 and 4.

## 3.2 OVERVIEW OF OUR APPROACH

At the highest level, the main algorithm relies on a simple yet powerful observation by Adil et al. (2019a), which is that the $\ell_p$ minimization problem we are attempting to solve supports iterative refinement. Adil et al. (2019a) show that having access to a weak solver which gives a constant factor multiplicative approximation to a mixed objective of $\ell_p$ and $\ell_2$ norms suffices to boost the multiplicative error to $1 + \epsilon$ while only making $\widetilde{O}_p(\log 1/\epsilon)$ calls to the solver. This reduces the entire difficulty of the problem to implementing the weak solver.

More precisely, starting with an initial solution (set to $\arg\min_{x:Ax=b} \|x\|_2$), we maintain $M^{(t)}$ as an upper bound for the function value gap, ie. $\|x^{(t)}\|_p^p - \|x^*\|_p^p \leq 16pM^{(t)}$. We show this invariant in Lemma E.2. In each iteration, the algorithm makes a call to a solver for the residual problem which approximates the function value progress $\|x\|_p^p - \|x - \Delta\|_p^p$ if we update the solution $x \leftarrow x - \Delta$. The residual solution tells us either the progress is too small, in which case we can improve the upperbound on the suboptimality gap by reducing $M^{(t)}$, or the progress is at least $\Omega\left(M^{(t)}\right)$, in which case we can perform the update and obtain a new solution. This new solution improves the function value gap by at least a factor $1 - \Omega\left(\frac{1}{p}\right)$, and thus the algorithm requires only $O\left(p \log \frac{\|x^{(0)}\|_p^p - \|x^*\|_p^p}{\epsilon \|x^*\|_p^p}\right)$ calls to the residual solver. We show this guarantee in Lemma E.2.

We give the pseudocode for the residual solver in Algorithm 4[3]. Prior works by Adil et al. (2019a;b; 2024) give algorithms for this solver either via a width-reduced multiplicative weights update method

---

[3]Note that while the residual solver takes as input the original matrix $A$ augmented with an extra row, the least squares problems required by the residual solver reduce to least squares problems involving only $A$, using the Sherman-Morrison formula. This guarantees that we only require a linear system solver for structured matrices of the form $A^\top D A$, for non-negative diagonal $D$.

---

**Algorithm 4** ResidualSolver$(p, A, b, \theta, M)$

---

**Input:** Matrix $A \in \mathbb{R}^{d \times n}$, vector $b \in \mathbb{R}^d$, target value $M$, weight $\theta$
**Output**: Vector $x$ such that $Ax = b$, $\|x\|_{2p} \leq 2M$ and $\langle \theta, x^2 \rangle \leq \min_{x:Ax=b} \|x^2\|_p + \langle \theta, x^2 \rangle$
$\qquad$ or approximate infeasibility certificate $r$, $\|r\|_q = 1$.

**if** $p \leq \frac{\log n}{\log n - 1}$ **then**
$\qquad r = \frac{1}{n^{\frac{1}{q}}}$; $\widehat{x} = \arg\min_{x:Ax=b} \langle r + \theta, x^2 \rangle$
$\qquad$ **if** $\|\widehat{x}\|_{2p} \leq 2M$ **then** return $\widehat{x}$ **else return** $r$ **end if**
**else**
$\qquad t = 0, r^{(0)} = \frac{2q-1}{2qn^{\frac{1}{q}}}, t' = 0, s^{(t')} = 0$
$\qquad$ **while** $\left\| (r^{(t)})^q \right\|_1 \leq 1$
$\qquad\qquad x^{(t)} = \arg\min_{x:Ax=b} \langle r^{(t)} + \theta, x^2 \rangle$
$\qquad\qquad \gamma_i^{(t)} = \begin{cases} \frac{x_i^2 \|r\|_q^{q-1}}{M^2 r_i^{q-1}} & \text{if } \frac{x_i^2 \|r\|_q^{q-1}}{r_i^{q-1}} \geq 2M^2 \\ 1 & \text{otherwise} \end{cases}$, for all $i$
$\qquad\qquad \alpha_i^{(t)} = \left( \gamma_i^{(t)} \right)^{1/q}$
$\qquad\qquad$ **if** $\alpha^{(t)} = 1$ **then return** $x^{(t)}$ **end if** $\qquad\qquad\qquad\qquad\qquad$ ▷ *Case 1*
$\qquad\qquad r^{(t+1)} = \alpha^{(t)} \cdot r^{(t)}$
$\qquad\qquad$ **if** $\alpha^{(t)} \leq n^{\frac{2}{2q+1}}$ **then** $s^{(t'+1)} = s^{(t')} + x^{(t)}; t' = t' + 1$ **end if**
$\qquad\qquad$ **if** $t' > 0$ **and** $\left\| s^{(t')}/t' \right\|_{2p} \leq 2M$ **then return** $s^{(t')}/t'$ **end if** $\qquad$ ▷ *Case 2*
$\qquad\qquad t = t + 1$
$\qquad$ **end while**
**end if**
**return** $r^{(t)}$ $\qquad\qquad\qquad\qquad\qquad\qquad\qquad\qquad\qquad\qquad\qquad\qquad\qquad\qquad$ ▷ *Case 3*

---

which achieves the state-of-the-art theoretical runtime but does not support a practical implementation or via a practical IRLS method with suboptimal theoretical guarantee. In contrast, we build on ideas from the low precision IRLS solver we have shown in the previous section and design a new IRLS algorithm that attains the best of both worlds.

Our residual solver outputs an approximate solution to a constant factor to the objective of the form

$$\min_{x:Ax=b} \|x^2\|_p + \langle \theta, x^2 \rangle \tag{4}$$

for $p \geq 1$ and a positive weight vector $\theta \in \mathbb{R}^n$. We also start with the dual formulation of the problem

$$(4) = \min_{x:Ax=b} \max_{r:\|r\|_q=1} \langle r, x^2 \rangle + \langle \theta, x^2 \rangle = \max_{r \geq 0:\|r\|_q=1} \min_{x:Ax=b} \langle r + \theta, x^2 \rangle = \max_{r \geq 0} \mathcal{E}\left( \frac{r}{\|r\|_q} + \theta \right),$$

where $q$ is the dual norm to $p$ and $\mathcal{E}(r + \theta) = \min_{x:Ax=b} \langle r + \theta, x^2 \rangle$. Given a target $M$, our goal is to find a primal solution $x$ that satisfies $\|x\|_{2p} \leq 2M$ and $\langle \theta, x^2 \rangle \leq \min_{x:Ax=b} \|x^2\|_p + \langle \theta, x^2 \rangle$ or a dual solution $r \in \mathbb{R}^n$ (infeasibility certificate) which can certify that $\min_{x:Ax=b} \|x\|_{2p}^2 \geq \mathcal{E}\left( \frac{r}{\|r\|_q} + \theta \right) \geq \frac{M^2}{2\kappa}$, where $\kappa$ is a value set as shown in Algorithm 3.

We distinguish between two regimes: when $p$ is sufficiently small, $1 \leq p \leq \frac{\log n}{\log n - 1}$ for which we will show that we can obtain a solution by $O(1)$ calls to the linear solver, and when $p > \frac{\log n}{\log n - 1}$, to which we need to pay more attention. In the latter case, similarly to Algorithm 2, we want to maintain the invariant

$$\frac{\mathcal{E}(r^{(t+1)} + \theta) - \mathcal{E}(r^{(t)} + \theta)}{\|r^{(t+1)}\|_q - \|r^{(t)}\|_q} \geq M^2.$$

Notice the differences between this objective and the problem $\min_{x:Ax=b} \|x^2\|_p$ which we solve in the previous section. The $\ell_2$ term $\langle \theta, x^2 \rangle$ makes this objective no longer scale-free. However, this $\ell_2$

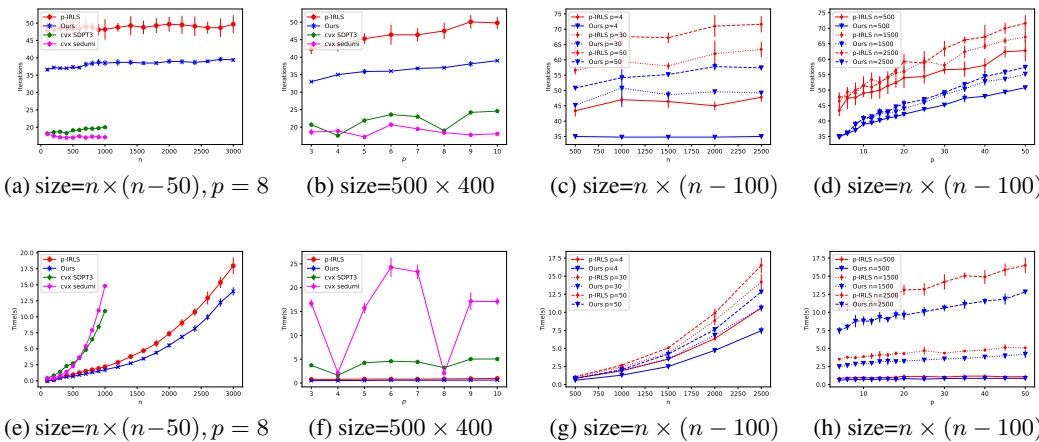

Figure 1: Performance on random matrices: $\min \|Ax - b\|_p^p$ with $\epsilon = 10^{-10}$. We compare our algorithm with CVX using SDPT3 and SeDuMi solvers and $p$-IRLS by Adil et al. (2019b). Figures (a),(b),(e),(f) plot the average and standard deviation of number of iterations and time taken by the solvers to find a solution over 10 runs. Figures (c),(d),(g),(h) measure over 5 runs.

term does not affect the lower bound $\sum_i r_i^{(t)} \left(x_i^{(t)}\right)^2 \left(1 - \frac{r_i^{(t)}}{r_i^{(t+1)}}\right)$ in the change in the objective

(eq. (2)); thus it suffices to maintain $\dfrac{\sum_i r_i^{(t)} \left(x_i^{(t)}\right)^2 \left(1 - \frac{r_i^{(t)}}{r_i^{(t+1)}}\right)}{\left\|r^{(t+1)}\right\|_q - \left\|r^{(t)}\right\|_q} \geq M^2$ in order to guarantee the

invariant $\frac{\mathcal{E}(r^{(t+1)} + \theta) - \mathcal{E}(r^{(t)} + \theta)}{\left\|r^{(t+1)}\right\|_q - \left\|r^{(t)}\right\|_q} \geq M^2$. At the same time, if we maintain $\|r\|_q \leq 1$, we can show that if the algorithm outputs a primal solution $x$, the $\ell_2$ term $\langle \theta, x^2 \rangle \leq \min_{x:Ax=b} \left\|x^2\right\|_p + \langle \theta, x^2 \rangle$. This requires us to initialize $r$ with sufficiently small $\|r\|_q$. Algorithm 4 then follows similarly to Algorithm 2, with the note that it suffices to obtain only a constant approximation. We give the correctness and convergence of Algorithm 4 in Lemma E.1 whose proof is based on the same idea as the analysis for Algorithm 2.

The complete analysis of our algorithm is provided in Appendix E.

## 4 EXPERIMENTAL EVALUATION

**On synthetic data.** We follow the experimental setup in Adil et al. (2019b), and build on the provided code[4]. We evaluate the performance of our high-precision Algorithm 3 on the problem $\min \|Ax - b\|_p^p$ on two types of instances: **(1)** Random matrices. The entries of $A$ and $b$ are generated uniformly at randomly between 0 and 1; **(2)** Random graphs. We use the procedure in Adil et al. (2019b) to generate random graphs and the corresponding $A$ and $b$. The generated graph is a weighted graph, where the vertices are generated by choosing a point in $[0, 1]^{10}$ uniformly at random, each vertex is connected to the 10 nearest neighbors. The edge weights are generated by a gaussian type function (by Flores-Calder-Lerman). $k$ (around 10) nodes are labeled in $[0, 1]$ and let $g$ be the label vector. Let $B$ be the edge-vertex adjacency matrix, $W$ be the diagonal matrix with edge weights. We generate $A = W^{1/p}B, b = -B[:, n : n + k]g$.

We vary $p$ and the size of the matrices and graphs, while keeping the error $\epsilon = 10^{-10}$. All implementations were done on MATLAB 2024a on a MacBook Pro M2 with 16GB RAM. We measure the number of iterations and running time for each algorithm and report them in Figures 1-2. In the appendix, we provide additional experimental results when $1 < p < 2$ and when $\epsilon$ varies.

---

[4]The code is available at `https://github.com/fast-algos/pIRLS`

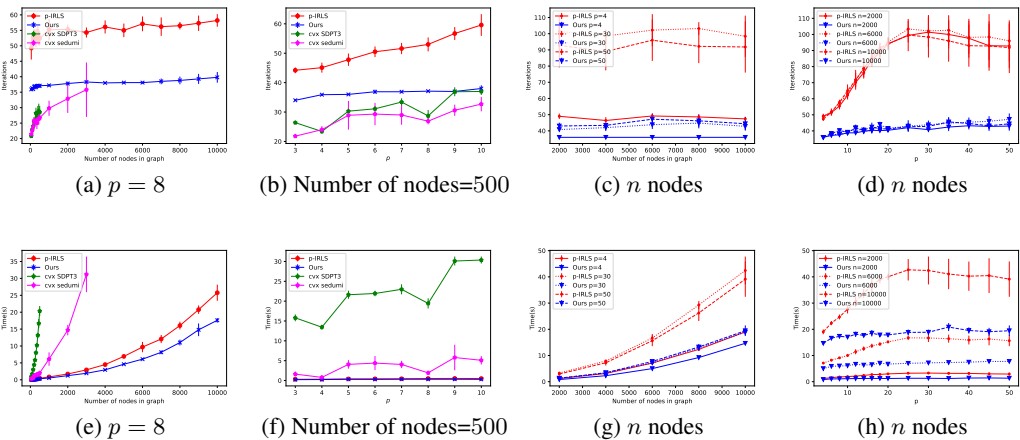

Figure 2: Performance on random graph instances: $\min \|Ax - b\|_p^p$ with $\epsilon = 10^{-10}$. We compare our algorithm with CVX using SDPT3 and SeDuMi solvers and $p$-IRLS by Adil et al. (2019b). Figures (a),(b),(e),(f) measure over 10 runs. Figures (c),(d),(g),(h) measure over 5 runs.

Table 1: Performance of our algorithm against $p$-IRLS on six real-world datasets for $p = 8$, $\epsilon = 10^{-10}$.

| | | CT slices Graf et al. (2011) | KEGG Metabolic Naeem and Asghar (2011) | Power Consumption Hebrail and Berard (2006) | Buzz in Social Media Kawala et al. (2013) | Protein Property Rana (2013) | Song Year Prediction Bertin-Mahieux (2011) |
|---|---|---|---|---|---|---|---|
| | Size | 48150 ×385 | 57248 ×27 | 1844352 ×11 | 524925 ×77 | 41157×9 | 463811 ×90 |
| no. | $p$-IRLS | 48 | 50 | 45 | 50 | 44 | 45 |
| iters | Ours | 36 | 42 | 36 | 42 | 36 | 36 |
| time | $p$-IRLS | 14.3 | 2.5 | 32. | 28. | 1.6 | 22.5 |
| (s) | Ours | 9.2 | 1.7 | 15.7 | 18.1 | 1.1 | 13.3 |

**On real-world datasets.** We test our algorithm against $p$-IRLS on six regression datasets from the UCI repository. CVX has excessive runtime and hence is excluded from the comparison. Results are provided in Table 1.

*Remark* 4.1. Regarding the correctness of the algorithm, we use the output by CVX as the baseline. In all experiments, our algorithm has error within the $\epsilon$ margin compared with the objective value of the CVX solution (see appendix).

On smaller instances, we compare our algorithm with CVX using SDPT3 and Sedumi solvers and the $p$-IRLS algorithm by Adil et al. (2019b). While CVX solvers generally need fewer iterations to find a solution, they are significantly slower on all instances than our algorithm and $p$-IRLS. Our algorithm also significantly outperforms $p$-IRLS in both the number of iterations (calls to a linear system solver) and running time. When the size of the problem and the value of $p$ increases, the gap between our algorithm and $p$-IRLS also increases. On average, our algorithm is 1-2.6 times faster than $p$-IRLS.

ACKNOWLEDGEMENT

AE was supported in part by an Alfred P. Sloan Research Fellowship. AV was partially supported by the French Agence Nationale de la Recherche (ANR) under grant ANR-21-CE48-0016 (project COMCOPT).

## REPRODUCIBILITY STATEMENT

For the reproducibility purpose, we submitted the source code in the supplementary material. We included the MATLAB implementation by Adil et al. (2019b).

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

## A    PROPERTY OF THE ENERGY FUNCTION

We recall the definition of energy function and its properties used in the algorithms.

**Definition A.1.** (Energy function). Given a vector $r \in \mathbb{R}^n_+$, we let the electrical energy be $\mathcal{E}(r) = \min_{x:Ax=b}\langle r, x^2\rangle$.

**Lemma A.1.** *(Computing the energy minimizer) Given $b \in \mathbb{R}^d$ and $r \in \mathbb{R}^n_+$, the least squares problem $\min_{x:Ax=b}\langle r, x^2\rangle$ can be solved by evaluating $x = \mathbb{D}(r)^{-1}A^\top \left(A\mathbb{D}(r)^{-1}A^\top\right)^+ b$, where $\mathbb{D}(r)$ is the diagonal matrix whose entries are given by $r$.*

The following lemma gives us a lower bound on the increase in electrical energy when we increase $r$.

**Lemma A.2.** *Given $r' \geq r$ and letting $x = \arg\min_{x:Ax=b}\langle r, x^2\rangle$, one has that*

$$\mathcal{E}(r') - \mathcal{E}(r) \geq \sum_i r_i x_i^2 \left(1 - \frac{r_i}{r_i'}\right).$$

*Proof.* This inequality follows from the standard lower bound for $\mathcal{E}(r') - \mathcal{E}(r)$, which the reader can find in Ene and Vladu (2019). $\square$

## B    REDUCING GENERAL REGRESSION PROBLEMS TO THE AFFINE-CONSTRAINED VERSION

In this section we show that the affine constrained version of the problem we consider is in full generality. Formally, we show that any $\ell_p$ regression problem of the form $\min_{Ax=b}\|Nx - v\|_p$ can be reduced to the form we consider.

**Lemma B.1.** *Let $A \in \mathbb{R}^{s \times n}, b \in \mathbb{R}^s, N \in \mathbb{R}^{m \times n}, v \in \mathbb{R}^m$ and consider the optimization objective $\min_{Ax=b} \|Nx - v\|_p$. Let $\begin{bmatrix} x \\ z \end{bmatrix}$ be a $(1 + \varepsilon)$ approximate solution to the affine-constrained regression problem*

$$\min_{\begin{bmatrix} N & -I_{m \times m} \\ A & 0_{s \times m} \end{bmatrix} \begin{bmatrix} x \\ z \end{bmatrix} = \begin{bmatrix} v \\ b \end{bmatrix}} \|z\|_p .$$

*Then $x$ is a $(1 + \varepsilon)$ approximate solution to the original objective. Furthermore, each least squares subproblem can be solved using two calls to a linear system solver for $N^\top R N$, and one call to a linear system solver for $A \left( N^\top R N \right)^+ A^\top$.*

*Proof.* We augment the dimension of the iterate by introducing $m$ additional variables encoded in a vector $z \in \mathbb{R}^m$. Hence one can equivalently enforce the constraints

$$Nx - z = v$$
$$Ax = b$$

and simply seek to minimize $\|z\|_p$ instead of $\|Ax - b\|_p$, which is the suitable formulation required by our solver. Note that while we do not have any weights on the $x$ iterate, the analysis goes through normally, since in fact it tolerates solving a more general weighted $\ell_p$ regression problem.

To solve the corresponding least squares problem, we need to compute

$$\min_{Ax=b} \frac{1}{2} \left\langle r, (Nx - v)^2 \right\rangle = \min_{Ax=b} \frac{1}{2} x^\top N^\top R N x - \left\langle N^\top R v, x \right\rangle + \frac{1}{2} v^\top R v$$

$$= \max_y \min_x \frac{1}{2} x^\top N^\top R N x - \left\langle N^\top R v, x \right\rangle + \frac{1}{2} v^\top R v + \langle b - Ax, y \rangle$$

$$= \max_y \left( \langle b, y \rangle + \min_x \frac{1}{2} x^\top N^\top R N x - \left\langle N^\top R v + A^\top y, x \right\rangle \right) - \frac{1}{2} v^\top R v .$$

where $R$ is the diagonal matrix whose entries are given by $r$. The inner problem is minimized at

$$x = \left( N^\top R N \right)^+ \left( N^\top R v + A^\top y \right) ,$$

which simplifies the problem to

$$\max_y \langle b, y \rangle - \frac{1}{2} \left( N^\top R v + A^\top y \right)^\top \left( N^\top R N \right)^+ \left( N^\top R v + A^\top y \right) - \frac{1}{2} v^\top R v$$

$$= \max_y \left\langle b - A \left( N^\top R N \right)^+ N^\top R v, y \right\rangle - \frac{1}{2} y^\top A \left( N^\top R N \right)^+ A^\top y$$

$$- \frac{1}{2} v^\top R N \left( N^\top R N \right)^+ N^\top R v - \frac{1}{2} v^\top R v ,$$

which is maximized at

$$y = \left( A \left( N^\top R N \right)^+ A^\top \right)^+ \left( b - A \left( N^\top R N \right)^+ N^\top R v \right) ,$$

so

$$x = \left( N^\top R N \right)^+ N^\top R v + \left( N^\top R N \right)^+ A^\top \left( A \left( N^\top R N \right)^+ A^\top \right)^+ \left( b - N \left( N^\top R N \right)^+ N^\top R v \right)$$

$$= \left( N^\top R N \right)^+ \left( N^\top R v + A^\top \left( A \left( N^\top R N \right)^+ A^\top \right)^+ \left( b - A \left( N^\top R N \right)^+ N^\top R v \right) \right) .$$

We observer that to execute this step we require two calls to a solver for $N^\top R N$, and one call to a solver for $A \left( N^\top R N \right)^+ A^\top$. $\qquad \square$

## C  Solving $\ell_p$ Regression for $1 \leq p < 2$

In this section we show that while our solvers are defined for $\ell_p$ regression when $p \geq 2$, they also provide solutions $\ell_q$ regression for $1 \leq q < 2$. This follows directly from exploiting duality. See Adil et al. (2019a), section 7.2 for a proof detailed. Here we briefly explain why this is the case. Let $p, q$ such that $\frac{1}{p} + \frac{1}{q} = 1$, $1 \leq q < 2$, and consider the $\ell_q$ regression problem, along with its dual

$$\min_{x : Ax = b} \|x\|_q = \max_{\|A^\top y\|_p \leq 1} \langle b, y \rangle \,.$$

We can use our solver to provide a high precision solution to the dual maximization problem, which we then show can be used to read off a primal nearly optimal solution. Indeed, we can equivalently solve

$$\min_{\langle b, y \rangle = 1} \left\| A^\top y \right\|_p$$

to high precision $\varepsilon = \frac{1}{n^{O(1)}}$, based on which we construct the nearly-feasible primal solution

$$x = \frac{\langle b, y \rangle}{\|A^\top y\|_p^p} \cdot \left( A^\top y \right)^{p-1} \,.$$

To see why this is a good solution, let us assume that we achieve exact gradient optimality for $y$, which means that for some scalar $\lambda$,

$$A \left( A^\top y \right)^{p-1} = b \cdot \lambda \,. \tag{5}$$

First let us verify that $x$ is feasible. Using (5) we see that:

$$Ax = A \left( \frac{\langle b, y \rangle}{\|A^\top y\|_p^p} \cdot \left( A^\top y \right)^{p-1} \right) = \frac{\langle b, y \rangle}{\|A^\top y\|_p^p} \cdot A \left( A^\top y \right)^{p-1} = \left( \frac{\langle b, y \rangle}{\|A^\top y\|_p^p} \cdot \lambda \right) \cdot b \,.$$

Additionally we can also use (5) again to obtain that

$$\left\| A^\top y \right\|_p^p = \left\langle y, A \left( A^\top y \right)^{p-1} \right\rangle = \langle y, b \rangle \cdot \lambda \,,$$

which allows us to conclude that

$$Ax = b \,,$$

so $x$ is feasible. Finally, we can measure the duality gap by calculating

$$\|x\|_q = \frac{1}{\lambda} \left\| \left( A^\top y \right)^{p-1} \right\|_q = \frac{1}{\lambda} \cdot \left( \sum \left( A^\top y \right)^{(p-1)\frac{p}{p-1}} \right)^{\frac{p-1}{p}} = \frac{1}{\lambda} \left\| A^\top y \right\|_p^{p-1}$$

$$= \frac{\langle y, b \rangle}{\|A^\top y\|_p^p} \cdot \left\| A^\top y \right\|_p^{p-1} = \frac{\langle y, b \rangle}{\|A^\top y\|_p} \,,$$

which certifies optimality for $b$. While in general we do not solve the dual problem exactly, which yields a slight violation in the demand for the primal iterate $x$, this can be fixed by adding to $x$ a flow $\widetilde{x} = A^\top \left( AA^\top \right)^+ (b - Ax)$ that routes the residual demand. This affects the $\ell_q$ norm only slightly since the residual demand is guaranteed to be very small due to the near-optimality of the dual problem. Then we can proceed to bounding the duality gap by following the argument sketched above, while also carrying the polynomially small error through the calculation. We refer the reader to Adil et al. (2019a) for the detailed error analysis. We have the following theorem.

**Theorem C.1.** *For any $1 < p \leq 2$, there is an iterative algorithm for the $\ell_p$ regression problem $\min_{x \in \mathbb{R}^n : Ax = b} \|x\|_p$ that solves $O\left( q^2 \log n \log \left( \frac{n}{\epsilon} \right) \right)$ subproblems, each of which makes $O\left( n^{\frac{q-2}{3q-2}} \right)$ calls to solve a linear system of the form $\widetilde{A} D \widetilde{A}^\top \phi = z$, where $q = \frac{p}{p-1}$, $D$ is an arbitrary non-negative diagonal matrix, $\widetilde{A}$ is a matrix obtained from $A$ by appending a single row, and $z$ is a vector obtained from the all-zero vector by appending a single non-zero coordinate.*

# D    PROOF OF THEOREM 1.1

*Proof of Lemma 2.1.* First we show (3).

$$\frac{1}{\left\|r^{(t+1)}\right\|_q - \left\|r^{(t)}\right\|_q} \geq \frac{q\left\|r^{(t)}\right\|_q^{q-1}}{\left\|r^{(t+1)}\right\|_q^q - \left\|r^{(t)}\right\|_q^q}.$$

This is equivalent to show

$$\left\|r^{(t+1)}\right\|_q^q + (q-1)\left\|r^{(t)}\right\|_q^q \geq q\left\|r^{(t+1)}\right\|_q \left\|r^{(t)}\right\|_q^{q-1}$$

which can easily be obtained from AM-GM inequality.

Using (3) and Lemma A.2 we have

$$\frac{\mathcal{E}(r^{(t+1)}) - \mathcal{E}(r^{(t)})}{\left\|r^{(t+1)}\right\|_q - \left\|r^{(t)}\right\|_q} \geq \frac{q\left\|r^{(t)}\right\|_q^{q-1}\left(\sum_i r_i^{(t)}\left(x_i^{(t)}\right)^2\left(1 - \frac{r_i^{(t)}}{r_i^{(t+1)}}\right)\right)}{\sum_i \left(r_i^{(t+1)}\right)^q - \left(r_i^{(t)}\right)^q}$$

$$= \frac{q\left\|r^{(t)}\right\|_q^{q-1}\left(\sum_{i,\alpha_i^{(t)}>1} r_i^{(t)}\left(x_i^{(t)}\right)^2\left(1 - \frac{r_i^{(t)}}{r_i^{(t+1)}}\right)\right)}{\sum_{i,\alpha_i^{(t)}>1}\left(r_i^{(t+1)}\right)^q - \left(r_i^{(t)}\right)^q}.$$

For $i$ such that $\alpha_i^{(t)} > 1$, we have $r_i^{(t+1)} = \alpha_i^{(t)} r_i^{(t)}$, thus

$$\frac{q\left\|r^{(t)}\right\|_q^{q-1} r_i^{(t)}\left(x_i^{(t)}\right)^2\left(1 - \frac{r_i^{(t)}}{r_i^{(t+1)}}\right)}{\left(r_i^{(t+1)}\right)^q - \left(r_i^{(t)}\right)^q} = \frac{\left\|r^{(t)}\right\|_q^{q-1}\left(x_i^{(t)}\right)^2}{\left(r_i^{(t)}\right)^{q-1}} \cdot \frac{q\left(1 - \frac{1}{\alpha_i^{(t)}}\right)}{\left(\alpha_i^{(t)}\right)^q - 1}$$

$$\geq \gamma_i^{(t)} M^2 \cdot \frac{1}{\left(\alpha_i^{(t)}\right)^q}$$

$$= M^2,$$

where the first inequality is due to $\frac{q(\alpha-1)}{\alpha(\alpha^q-1)} \geq \frac{1}{\alpha^q}$, for $\alpha > 1$. We can then obtain the desired conclusion from here. $\qquad\square$

*Proof of Lemma 2.2.* If

$$\left\|\|r\|_q^{q-1} \cdot \frac{x^2}{r^{q-1}}\right\|_\infty \leq (1+\epsilon)M^2,$$

for all $i$ we have

$$x_i^2 \leq (1+\epsilon)^2 M^2 \frac{r_i^{q-1}}{\|r\|_q^{q-1}},$$

which gives

$$x_i^{2p} \leq (1+\epsilon)^{2p} M^{2p} \frac{r_i^q}{\|r\|_q^q},$$

We obtain

$$\|x\|_{2p}^{2p} \leq (1+\epsilon)^{2p} M^{2p},$$

as needed. $\qquad\square$

*Proof of Lemma 2.3.* We have that

$$
\frac{\mathcal{E}(r^{(T)})}{\left\|r^{(T)}\right\|_q} = \frac{\mathcal{E}(r^{(0)}) + \sum_{t=0}^{T-1}\left(\mathcal{E}(r^{(t+1)}) - \mathcal{E}(r^{(t)})\right)}{\left\|r^{(T)}\right\|_q}
$$

$$
\geq \frac{\mathcal{E}(r^{(0)}) + \sum_{t=0}^{T-1}\left(\left\|r^{(t+1)}\right\|_q - \left\|r^{(t)}\right\|_q\right)\cdot M^2}{\left\|r^{(T)}\right\|_q} \quad \text{(due to the invariant)}
$$

$$
\geq \frac{\left(\left\|r^{(T)}\right\|_q - 1\right)\cdot M^2}{\left\|r^{(T)}\right\|_q} = M^2\cdot\left(1 - \frac{1}{\left\|r^{(T)}\right\|_q}\right)
$$

$$
\geq M^2\cdot(1-\epsilon) \quad \text{(since } \left\|r^{(T)}\right\|_q \geq \frac{1}{\epsilon}\text{)}
$$

$$
\geq \frac{M^2}{(1+\epsilon)^2}.
$$

$\square$

*Proof of Lemma 2.4.* Suppose the contrary. Then we claim that the perturbations that scale the dual solution by $\geq S$ will have increased it a lot to the point where $\|r\|_q^q \geq \frac{1}{\epsilon^q}$. Indeed, since $r$ is initialized to $\frac{1}{n^{1/q}}$, in the worst case each perturbation in $T_{hi}$ touches a different coordinate $i$. Therefore this establishes a lower bound of $T_{hi}\cdot\frac{S^q}{n}$ on $\|r\|_q^q$. As this must be at most $\frac{1}{\epsilon^q}$, since otherwise we obtained a good solution per Lemma 2.3, we obtain the conclusion. $\square$

Before showing the proof of Lemma 2.5, we claim that we can either look at the history produced in $T_{lo}$ and obtain an approximately feasible solution, or a single coordinate of $r$ must have increased a lot.

**Lemma D.1.** *Consider the set of iterates $(r^{(t)}, x^{(t)})$ used for the iterates in $T_{lo}$. If*

$$
\left\|\frac{1}{T_{lo}}\sum_{t\in T_{lo}} x^{(t)}\right\|_{2p} > M(1+\epsilon)
$$

*then there exists a coordinate $i$ for which*

$$
\sum_{t\in T_{lo}:\alpha_i^{(t)}>1}\sqrt{\alpha_i^{(t)}} \geq \frac{T_{lo}\epsilon^{\frac{q+1}{2}}}{2}.
$$

*Proof.* Suppose that

$$
\left\|\frac{1}{T_{lo}}\sum_{t\in T_{lo}} x^{(t)}\right\|_{2p} > M(1+\epsilon)
$$

Note that by the update rule,

$$
\frac{x_i^{(t)}}{M} \leq (1+\epsilon)^{\frac{1}{2}}\sqrt{\frac{\left(r_i^{(t)}\right)^{q-1}}{\left\|r^{(t)}\right\|_q^{q-1}}} + \mathbf{1}_{\alpha_i>1}\sqrt{\frac{\alpha_i^{(t)q}\left(r_i^{(t)}\right)^{q-1}}{\left\|r^{(t)}\right\|_q^{q-1}}}
$$

$$
\leq \left(1+\frac{\epsilon}{2}\right)\sqrt{\frac{\left(r_i^{(t)}\right)^{q-1}}{\left\|r^{(t)}\right\|_q^{q-1}}} + \mathbf{1}_{\alpha_i>1}\sqrt{\frac{\alpha_i^{(t)q}\left(r_i^{(t)}\right)^{q-1}}{\left\|r^{(t)}\right\|_q^{q-1}}}
$$

Hence we can write

$$
\left\|\sum_{t\in T_{lo}}\frac{x^{(t)}}{M}\right\|_{2p} \leq \left\|\left(1+\frac{\epsilon}{2}\right)\sum_{t\in T_{lo}}\sqrt{\frac{\left(r^{(t)}\right)^{q-1}}{\left\|r^{(t)}\right\|_q^{q-1}}} + \overrightarrow{\left(\sum_{t\in T_{lo},\alpha_i^{(t)}>1}\sqrt{\frac{\alpha_i^{(t)q}\left(r_i^{(t)}\right)^{q-1}}{\left\|r^{(t)}\right\|_q^{q-1}}}\right)_i}\right\|_{2p}
$$

$$\leq \left(1 + \frac{\epsilon}{2}\right) \sum_{t \in T_{lo}} \left\| \sqrt{\frac{\left(r^{(t)}\right)^{q-1}}{\left\| r^{(t)} \right\|_q^{q-1}}} \right\|_{2p} + \left\| \overrightarrow{\left( \sum_{t \in T_{lo}, \alpha_i^{(t)} > 1} \sqrt{\frac{\alpha_i^{(t)q} \left(r_i^{(t)}\right)^{q-1}}{\left\| r^{(t)} \right\|_q^{q-1}}} \right)_i} \right\|_{2p}$$

(by triangle inequality)

$$= \left(1 + \frac{\epsilon}{2}\right) T_{lo} + \left\| \overrightarrow{\left( \sum_{t \in T_{lo}, \alpha_i^{(t)} > 1} \sqrt{\frac{\alpha_i^{(t)q} \left(r_i^{(t)}\right)^{q-1}}{\left\| r^{(t)} \right\|_q^{q-1}}} \right)_i} \right\|_{2p}.$$

We obtain

$$\left\| \overrightarrow{\left( \sum_{t \in T_{lo}, \alpha_i^{(t)} > 1} \sqrt{\frac{\alpha_i^{(t)q} \left(r_i^{(t)}\right)^{q-1}}{\left\| r^{(t)} \right\|_q^{q-1}}} \right)_i} \right\|_{2p} \geq \frac{\epsilon}{2} T_{lo}$$

On the other hand, we have

$$\sum_i \left( \sum_{t \in T_{lo}, \alpha_i^{(t)} > 1} \sqrt{\frac{\alpha_i^{(t)q} \left(r_i^{(t)}\right)^{q-1}}{\left\| r^{(t)} \right\|_q^{q-1}}} \right)^{2p} = \sum_i \left( \sum_{t \in T_{lo}, \alpha_i^{(t)} > 1} \sqrt{\frac{\alpha_i^{(t)} \left(r_i^{(t+1)}\right)^{q-1}}{\left\| r^{(t)} \right\|_q^{q-1}}} \right)^{2p}$$

$$\leq \sum_i \left(r_i^{(T)}\right)^q \left( \sum_{t \in T_{lo}, \alpha_i^{(t)} > 1} \sqrt{\alpha_i^{(t)}} \right)^{2p} \leq \left\| r^{(T)} \right\|_q^q \max_i \left( \sum_{t \in T_{lo}, \alpha_i^{(t)} > 1} \sqrt{\alpha_i^{(t)}} \right)^{2p}$$

$$\leq \frac{1}{\epsilon^q} \max_i \left( \sum_{t \in T_{lo}, \alpha_i^{(t)} > 1} \sqrt{\alpha_i^{(t)}} \right)^{2p}$$

$\square$

Therefore there exists $i$ such that

$$\left( \sum_{t \in T_{lo}, \alpha_i^{(t)} > 1} \sqrt{\alpha_i^{(t)}} \right)^{2p} \geq \left( \frac{\epsilon T}{2} \right)^{2p} \epsilon^q,$$

which gives us

$$\sum_{t \in T_{lo}, \alpha_i^{(t)} > 1} \sqrt{\alpha_i^{(t)}} \geq \frac{T_{lo} \epsilon^{\frac{q+1}{2}}}{2}.$$

Now we show the proof of Lemma 2.5.

*Proof of Lemma 2.5.* From Lemma D.1 we know that there exists a coordinate $i$ for which

$$\sum_{t \in T_{lo}: \alpha_i^{(t)} > 1} \sqrt{\alpha_i^{(t)}} > \frac{T_{lo} \epsilon^{\frac{q+1}{2}}}{2}. \tag{6}$$

Furthermore by definition for all iterates in $T_{lo}$ we have that pointwise $(1 + \epsilon) \leq \left(\alpha_i^{(t)}\right)^q \leq S^q$. This enables us to lower bound the final value of $\left(r_i^{(T)}\right)^q$ which is a lower bound on $\left\| r^{(T)} \right\|_q^q$. More

precisely, we have

$$\left(r_i^{(T)}\right)^q \geq \left(r_i^{(0)}\right)^q \cdot \prod_{t \in T_{lo}: \alpha_i^{(t)} > 1} \left(\alpha_i^{(t)}\right)^q = \frac{1}{n} \cdot \prod_{t \in T_{lo}: \alpha_i^{(t)} > 1} \left(\alpha_i^{(t)}\right)^q. \tag{7}$$

Now we can proceed to lower bound this coodinate i.e. we lower bound the product in (7) using the lower bound we have in (6).

Intuitively, the worst case behavior i.e. slowest possible increase in $\left(r_i^{(T)}\right)^q$ is achieved in one of the two extreme cases:

(i) the $\alpha_i^{(t)}$ are all minimized i.e. $\left(\alpha_i^{(t)}\right)^q = (1 + \epsilon)$ in which case $\Theta\left(\frac{1}{\epsilon} \log\left(\frac{n}{\epsilon^q}\right)\right)$ such terms are sufficient to make their product $\geq \frac{n}{\epsilon^q}$, which means that we are done, since then we have $\left\|r^{(T)}\right\|_q^q \geq \left(r_i^{(T)}\right)^q \geq \frac{1}{\epsilon^q}$; so setting $\frac{T_{lo}\epsilon^{\frac{q+1}{2}}}{2} \geq \Theta\left((1 + \epsilon)^{\frac{1}{2q}} \frac{1}{\epsilon} \log\left(\frac{n}{\epsilon^q}\right)\right)$ i.e $T_{lo} \geq \Theta\left(\frac{1}{\epsilon^{\frac{q+3}{2}}} \log\left(\frac{n}{\epsilon^q}\right)\right)$ is sufficient to make this happen;

(ii) all the entries are maximized, i.e. $\alpha_i^{(t)} = S$ in which case we have that their product to power $q$ is at least $S^{\frac{qT_{lo}}{S^{1/2}} \frac{\epsilon^{\frac{q+1}{2}}}{2}} \geq \frac{n}{\epsilon^q}$, so if we set $\frac{qT_{lo}}{S^{1/2}} \frac{\epsilon^{\frac{q+1}{2}}}{2} \ln S \geq \log\left(\frac{n}{\epsilon^q}\right)$, ie., $T_{lo} = \Theta\left(\frac{S^{1/2}}{q \ln S} \frac{1}{\epsilon^{\frac{q+1}{2}}} \log\left(\frac{n}{\epsilon^q}\right)\right)$, we guarantee that the corresponding $r_i$ increases to a value larger than $\frac{1}{\epsilon^q}$. The fact that these two cases capture the slowest possible increase is shown in Lemma F.1.

Therefore we can set

$$T_{lo} = O\left(\left(\frac{1}{\epsilon} + \frac{S^{1/2}}{q \ln S}\right) \frac{1}{\epsilon^{\frac{q+1}{2}}} \log\left(\frac{n}{\epsilon^q}\right)\right).$$

$\square$

## E    PROOF OF THEOREM 1.2

First, we give guarantee for the subproblem solver (Algorithm 4, proof follows subsequently) .

**Lemma E.1.** *For $p \geq 1$, $\kappa = \begin{cases} 1 & \text{if } p \leq \frac{\log n}{\log n - 1} \\ q & \text{otherwise} \end{cases}$, Algorithm 4 either returns $x$ such that $Ax = b$, $\|x\|_{2p} \leq 2M$ and $\langle\theta, x^2\rangle \leq \min_{x:Ax=b} \|x^2\|_p + \langle\theta, x^2\rangle$ or certifies that $\min_{x:Ax=b} \|x^2\|_p + \langle\theta, x^2\rangle \geq \frac{M^2}{2\kappa}$ in $O\left(n^{\frac{1}{2q+1}}\right)$ calls to solve a linear system of the form $ADA^\top \phi = b$, where $D$ is an arbitrary non-negative diagonal matrix.*

The next lemma provides guarantees on the iterate progress in the main algorithm (Algorithm 3).

**Lemma E.2.** *For $p \geq 2$ $\kappa = \begin{cases} 1 & \text{if } p \leq \frac{2 \log n}{\log n - 1} \\ \frac{p}{p-2} & \text{otherwise} \end{cases}$, Algorithm 3 maintains that $\left\|x^{(t)}\right\|_p^p - \|x^*\|_p^p \leq 16pM^{(t)}$ and that if $x^{(t+1)} \neq x^{(t)}$ then*

$$\left\|x^{(t+1)}\right\|_p^p - \|x^*\|_p^p \leq \left(1 - \frac{1}{2^{13}p\kappa}\right)\left(\left\|x^{(t)}\right\|_p^p - \|x^*\|_p^p\right).$$

Finally, we show the proof of Theorem 1.2.

*Proof.* Algorithm 3 terminates when $M^{(t)} \leq \frac{\epsilon}{16p(1+\epsilon)} \left\|x^{(t)}\right\|_p^p$. This gives $\left\|x^{(t)}\right\|_p^p - \|x^*\|_p^p \leq \frac{\epsilon}{16p(1+\epsilon)} \left\|x^{(t)}\right\|_p^p$, which implies $\left\|x^{(t)}\right\|_p^p \leq (1 + \epsilon) \|x^*\|_p^p$ and thus $\left\|x^{(t)}\right\|_p \leq (1 + \epsilon) \|x^*\|_p$. Hence, $x^{(t)}$ is a $(1 + \epsilon)$ approximate solution. Since $\frac{\epsilon}{16p(1+\epsilon)} \left\|x^{(t)}\right\|_p^p \geq \frac{\epsilon}{16p(1+\epsilon)} \|x^*\|_p^p$, the number of times $M^{(t)}$ can be reduced is $O\left(\log \frac{\|x^{(0)}\|_p^p}{\epsilon \|x^*\|_p^p}\right) = O\left(p \log \frac{n}{\epsilon}\right)$. By Lemma E.2, the number of times the

iterate makes progress is $O\left(2^{13}p\kappa\log\frac{\|x^{(0)}\|_p^p - \|x^*\|_p^p}{\epsilon\|x^*\|_p^p}\right) = O\left(p^2\log n\log\frac{n}{\epsilon}\right)$ where $\kappa = O(\log n)$. Therefore the total number of calls to the subroutine solver is $O\left(p^2\log n\log\frac{n}{\epsilon}\right)$. By lemma E.1, the subroutine solver makes $O\left(n^{\frac{1}{2q+1}}\right) = O\left(n^{\frac{p-2}{3p-2}}\right)$ calls to a linear system solver. This concludes the proof. $\qquad\square$

## E.1 PROOF OF LEMMA E.1

We let $\mathcal{OPT} = \min_{x:Ax=b}\|x^2\|_p + \langle\theta, x^2\rangle$ and $x^* = \arg\min_{x:Ax=b}\|x^2\|_p + \langle\theta, x^2\rangle$. We consider two cases: when $p \leq \frac{\log n}{\log n - 1}$ and when $p > \frac{\log n}{\log n - 1}$. We will prove for each case using the following lemmas:

**Lemma E.3.** *For $1 \leq p \leq \frac{\log n}{\log n - 1}$, Algorithm 4 either returns $x$ such that $Ax = b$, $\|x\|_{2p} \leq 2M$ and $\langle\theta, x^2\rangle \leq \mathcal{OPT}$ or certifies that $\mathcal{OPT} \geq \frac{M^2}{2}$ in $O(1)$ call to solve a linear system.*

**Lemma E.4.** *For $p > \frac{\log n}{\log n - 1}$, Algorithm 4 either returns $x$ such that $Ax = b$, $\|x\|_{2p} \leq 2M$ and $\langle\theta, x^2\rangle \leq \mathcal{OPT}$ or certifies that $\mathcal{OPT} \geq \frac{M^2}{2q}$ in $O\left(n^{\frac{1}{2q+1}}\right)$ calls to solve a linear system.*

To start, we have the following lemma that controls the $\ell_2$ term in the objective

**Lemma E.5.** *For $r$ such that $\|r\|_q \leq 1$, suppose $x = \arg\min_{x:Ax=b}\langle r + \theta, x^2\rangle$. Then we have $\langle\theta, x^2\rangle \leq \mathcal{OPT}$.*

*Proof.* For $r$ with $\|r\|_q \leq 1$, we have

$$\langle\theta, x^2\rangle \leq \langle r + \theta, x^2\rangle \leq \langle r + \theta, (x^*)^2\rangle \quad \text{(by definition of } x\text{)}$$
$$\leq \|(x^*)^2\|_p + \langle\theta, (x^*)^2\rangle \leq \mathcal{OPT}.$$

$\qquad\square$

Now, let us turn to the first case when $1 \leq p \leq \frac{\log n}{\log n - 1}$. We give the proof for Lemma E.3.

*Proof of Lemma E.3.* When $1 \leq p \leq \frac{\log n}{\log n - 1}$, we have $q = \frac{p}{p-1} \geq \log n$. Algorithm 4 computes

$$\widehat{x} = \min_{x:Ax=b}\langle r + \theta, x^2\rangle$$

where $r_i = n^{-\frac{1}{q}}$ for all $i$.

Since $\|r\|_q = 1$, if $\|\widehat{x}\|_{2p} \leq 2M$, by Lemma E.5, we immediately have $\|\widehat{x}\|_{2p} \leq 2M$ and $\langle\theta, x^2\rangle \leq \mathcal{OPT}$.

Assume that $\|\widehat{x}\|_{2p} > 2M$. We have

$$\mathcal{OPT} = \|(x^*)^2\|_p + \langle\theta, (x^*)^2\rangle \geq \langle r, (x^*)^2\rangle + \langle\theta, (x^*)^2\rangle$$
$$= \langle\theta + r, (x^*)^2\rangle \geq \langle\theta + r, (\widehat{x})^2\rangle$$
$$\geq \frac{1}{n^{\frac{1}{q}}}\|\widehat{x}^2\|_1 \geq \frac{1}{n^{\frac{1}{q}}}\|\widehat{x}^2\|_p \qquad \text{(since } \|\widehat{x}^2\|_1 \geq \|\widehat{x}^2\|_p\text{)}$$
$$\geq \frac{1}{2}\|\widehat{x}\|_{2p}^2 \qquad \text{(since } q \geq \log n\text{)}$$
$$\geq 2M^2 \geq \frac{M^2}{2}.$$

$\qquad\square$

For the case when $p > \frac{\log n}{\log n - 1}$, the proof for Lemma E.4 follows similarly to the analysis of Algorithm 2. We proceed by showing the following invariant.

**Lemma E.6** (Invariant). *For all $t$, we have that if $\gamma^{(t)} \neq 1$ then $\frac{\mathcal{E}(r^{(t+1)}+\theta)-\mathcal{E}(r^{(t)}+\theta)}{\left\|r^{(t+1)}\right\|_q - \left\|r^{(t)}\right\|_q} \geq M^2$.*

*Proof.* Using Lemma A.2 we have

$$
\begin{aligned}
\frac{\mathcal{E}(r^{(t+1)}+\theta)-\mathcal{E}(r^{(t)}+\theta)}{\left\|r^{(t+1)}\right\|_q - \left\|r^{(t)}\right\|_q} &\geq \frac{q \cdot \left\|r^{(t)}\right\|_q^{q-1} \left(\sum_i \left(r_i^{(t)}+\theta_i\right)\left(x_i^{(t)}\right)^2 \left(1 - \frac{r_i^{(t)}+\theta_i}{r_i^{(t+1)}+\theta_i}\right)\right)}{\sum_i \left(r_i^{(t+1)}\right)^q - \left(r_i^{(t)}\right)^q} \\
&= \frac{q \cdot \left\|r^{(t)}\right\|_q^{q-1} \left(\sum_i \left(x_i^{(t)}\right)^2 \frac{r_i^{(t)}+\theta_i}{r_i^{(t+1)}+\theta_i} \left(r_i^{(t+1)} - r_i^{(t)}\right)\right)}{\sum_i \left(r_i^{(t+1)}\right)^q - \left(r_i^{(t)}\right)^q} \\
&\geq \frac{q \cdot \left\|r^{(t)}\right\|_q^{q-1} \left(\sum_i \left(x_i^{(t)}\right)^2 \frac{r_i^{(t)}}{r_i^{(t+1)}} \left(r_i^{(t+1)} - r_i^{(t)}\right)\right)}{\sum_i \left(r_i^{(t+1)}\right)^q - \left(r_i^{(t)}\right)^q} \\
&= \frac{q \cdot \left\|r^{(t)}\right\|_q^{q-1} \left(\sum_{i,\alpha_i^{(t)}>1} \left(x_i^{(t)}\right)^2 \frac{r_i^{(t)}}{r_i^{(t+1)}} \left(r_i^{(t+1)} - r_i^{(t)}\right)\right)}{\sum_{i,\alpha_i^{(t)}>} \left(r_i^{(t+1)}\right)^q - \left(r_i^{(t)}\right)^q},
\end{aligned}
$$

where in the second inequality we use $\frac{r_i^{(t)}+\theta_i}{r_i^{(t+1)}+\theta_i} \geq \frac{r_i^{(t)}}{r_i^{(t+1)}}$ for $r_i^{(t+1)} \geq r_i^{(t)}$, $\theta \geq 0$. For $i$ such that $\alpha_i^{(t)} > 1$, we have $r_i^{(t+1)} = \alpha_i^{(t)} r_i^{(t)}$, thus

$$
\begin{aligned}
\frac{q \cdot \left\|r^{(t)}\right\|_q^{q-1} \left(x_i^{(t)}\right)^2 \frac{r_i^{(t)}}{r_i^{(t+1)}} \left(r_i^{(t+1)} - r_i^{(t)}\right)}{\left(r_i^{(t+1)}\right)^q - \left(r_i^{(t)}\right)^q} &= \gamma_i^{(t)} M^2 \cdot \frac{q\left(1 - \frac{1}{\alpha_i^{(t)}}\right)}{\left(\alpha_i^{(t)}\right)^q - 1} \\
&\geq \gamma_i^{(t)} M^2 \cdot \frac{1}{\left(\alpha_i^{(t)}\right)^q} \\
&= M^2,
\end{aligned}
$$

where the first inequality is due to $\frac{q(\alpha-1)}{\alpha(\alpha^q-1)} \geq \frac{1}{\alpha^q}$, for $\alpha > 1$. We can then obtain the desired conclusion from here. $\qquad\square$

**Lemma E.7** (Case 1). *Let $r$ be a dual solution and $x = \arg\min_{\widehat{x}:A\widehat{x}=b}\langle r + \theta, \widehat{x}^2 \rangle$. If $\left\|\|r\|_q^{q-1} \cdot \frac{x^2}{r^{q-1}}\right\|_\infty \leq 2M$ then $\|x\|_{2p} \leq 2M$ and $\langle \theta, x^2 \rangle \leq \mathcal{OPT}$.*

*Proof.* If

$$
\left\|\|r\|_q^{q-1} \cdot \frac{x^2}{r^{q-1}}\right\|_\infty \leq 2M^2,
$$

for all $i$ we have

$$
x_i^2 \leq 4M^2 \frac{r_i^{q-1}}{\|r\|_q^{q-1}},
$$

which gives

$$
x_i^{2p} \leq 2^{2p} M^{2p} \frac{r_i^q}{\|r\|_q^q},
$$

We obtain

$$
\|x\|_{2p}^{2p} \leq 2^{2p} M^{2p},
$$

as needed. The second claim comes directly from Lemma E.5. $\qquad\square$

**Lemma E.8** (Case 3). *If the algorithm returns $r^{(T)}$, then $\mathcal{E}\left(\frac{r^{(T)}}{\left\|r^{(T)}\right\|_q} + \theta\right) \geq \frac{M^2}{2q}$.*

*Proof.* We have that

$$
\begin{aligned}
\frac{\mathcal{E}(r^{(T)} + \theta)}{\left\|r^{(T)}\right\|_q} &= \frac{\mathcal{E}(r^{(0)} + \theta) + \sum_{t=0}^{T-1}\left(\mathcal{E}(r^{(t+1)} + \theta) - \mathcal{E}(r^{(t)} + \theta)\right)}{\left\|r^{(T)}\right\|_q} \\
&\geq \frac{\sum_{t=0}^{T-1}\left(\left\|r^{(t+1)}\right\|_q - \left\|r^{(t)}\right\|_q\right) \cdot M^2}{\left\|r^{(T)}\right\|_q} \quad \text{(due to the invariant)} \\
&\geq \frac{\left(\left\|r^{(T)}\right\|_q - \left\|r^{(0)}\right\|_q\right) \cdot M^2}{\left\|r^{(T)}\right\|_q} \\
&= M^2 \cdot \left(1 - \frac{\frac{2q-1}{2q}}{\left\|r^{(T)}\right\|_q}\right) \quad \text{(since } \left\|r^{(0)}\right\|_q = \frac{2q-1}{2q}) \\
&= \frac{M^2}{2q} \quad \text{(since } \left\|r^{(T)}\right\|_q \geq 1).
\end{aligned}
$$

Finally since $\left\|r^{(T)}\right\|_q \geq 1$

$$
\mathcal{E}\left(\frac{r^{(T)}}{\left\|r^{(T)}\right\|_q} + \theta\right) \geq \frac{\mathcal{E}(r^{(T)} + \theta)}{\left\|r^{(T)}\right\|_q} \geq \frac{M^2}{2q}.
$$

$\square$

**Convergence Analysis**  We run the algorithm for $T$ iterations. The algorithm terminates if at any point it finds a solution $x$ that satisfies the desired bound (otherwise it is unable to further perturb the dual solution). Otherwise, we show that it must finish very fast.

Suppose we run it for $T = T_{hi} + T_{lo}$ iterations. Let the iterations in $T_{hi}$ correspond to those where at least a single $r_i$ was scaled by $\geq S = n^{\frac{2}{2q+1}}$. Let $T_{lo}$ be the remaining iterations.

**Lemma E.9.** *We have $T_{hi} \leq \frac{2n}{S^q}$.*

*Proof.* Suppose the contrary. Then we claim that these perturbations alone will have increased $r$ a lot to the point where $\|r\|_q^q \geq 1$. Indeed, let $r_i$ be the current value of coordinate $i$ and $r_i'$ be its value after being increased, and assume that $\frac{r_i'}{r_i} \geq S$. Since $r$ is initialized to $\frac{2q-1}{2q} \frac{1}{n^{1/q}}$, in the worst case each perturbation in $T_{hi}$ touches a different $i$. Therefore this establishes a lower bound of $T_{hi} \cdot \frac{S^q}{n}\left(\frac{2q-1}{2q}\right)^q \geq T_{hi} \cdot \frac{S^q}{2n}$ on $\|r\|_q^q$. As this must be at most 1, since otherwise we obtained a good solution per Lemma E.8, we obtain the conclusion. $\square$

Now we claim that we can either look at the history produced in $T_{lo}$ and obtain an approximately feasible solution, or a single coordinate $r_i$ must have increased a lot.

**Lemma E.10.** *Consider the set of iterates $(r^{(t)}, x^{(t)})$ used for the iterates in $T_{lo}$. If*

$$
\left\|\frac{1}{T_{lo}}\sum_{t\in T_{lo}} x^{(t)}\right\|_{2p} > 2M
$$

*then there exists a coordinate $i$ for which*

$$
\sum_{t\in T_{lo}:\alpha_i^{(t)}>1} \sqrt{\alpha_i^{(t)}} \geq \frac{T_{lo}}{4}.
$$

*Proof.* Suppose $\left\|\frac{1}{T_{lo}}\sum_{t\in T_{lo}}x^{(t)}\right\|_{2p} > 2M$. Note that by the update rule,

$$\frac{x_i^{(t)}}{M} \leq \sqrt{2}\sqrt{\frac{\left(r_i^{(t)}\right)^{q-1}}{\left\|r^{(t)}\right\|_q^{q-1}}} + \mathbf{1}_{\alpha_i>1}\sqrt{\frac{\alpha_i^{(t)q}\left(r_i^{(t)}\right)^{q-1}}{\left\|r^{(t)}\right\|_q^{q-1}}}$$

Hence we can write

$$\left\|\sum_{t\in T_{lo}}\frac{x^{(t)}}{M}\right\|_{2p} \leq \left\|\sqrt{2}\sum_{t\in T_{lo}}\sqrt{\frac{\left(r^{(t)}\right)^{q-1}}{\left\|r^{(t)}\right\|_q^{q-1}}} + \overrightarrow{\left(\sum_{t\in T_{lo},\alpha_i^{(t)}>1}\sqrt{\frac{\alpha_i^{(t)q}\left(r_i^{(t)}\right)^{q-1}}{\left\|r^{(t)}\right\|_q^{q-1}}}\right)_i}\right\|_{2p}$$

$$\leq \sqrt{2}\sum_{t\in T_{lo}}\left\|\sqrt{\frac{\left(r^{(t)}\right)^{q-1}}{\left\|r^{(t)}\right\|_q^{q-1}}}\right\|_{2p} + \left\|\overrightarrow{\left(\sum_{t\in T_{lo},\alpha_i^{(t)}>1}\sqrt{\frac{\alpha_i^{(t)q}\left(r_i^{(t)}\right)^{q-1}}{\left\|r^{(t)}\right\|_q^{q-1}}}\right)_i}\right\|_{2p}$$

(by triangle inequality)

$$= \sqrt{2}T_{lo} + \left\|\overrightarrow{\left(\sum_{t\in T_{lo},\alpha_i^{(t)}>1}\sqrt{\frac{\alpha_i^{(t)q}\left(r_i^{(t)}\right)^{q-1}}{\left\|r^{(t)}\right\|_q^{q-1}}}\right)_i}\right\|_{2p}.$$

We obtain

$$\left\|\overrightarrow{\left(\sum_{t\in T_{lo},\alpha_i^{(t)}>1}\sqrt{\frac{\alpha_i^{(t)q}\left(r_i^{(t)}\right)^{q-1}}{\left\|r^{(t)}\right\|_q^{q-1}}}\right)_i}\right\|_{2p} \geq \left(2-\sqrt{2}\right)T_{lo} \geq \frac{T_{lo}}{2}$$

On the other hand, we have

$$\sum_i\left(\sum_{t\in T_{lo},\alpha_i^{(t)}>1}\sqrt{\frac{\alpha_i^{(t)q}\left(r_i^{(t)}\right)^{q-1}}{\left\|r^{(t)}\right\|_q^{q-1}}}\right)^{2p} = \sum_i\left(\sum_{t\in T_{lo},\alpha_i^{(t)}>1}\sqrt{\frac{\alpha_i^{(t)}\left(r_i^{(t+1)}\right)^{q-1}}{\left\|r^{(t)}\right\|_q^{q-1}}}\right)^{2p}$$

$$\leq \sum_i\frac{\left(r_i^{(T)}\right)^q}{\left\|r^{(0)}\right\|_q^q}\left(\sum_{t\in T_{lo},\alpha_i^{(t)}>1}\sqrt{\alpha_i^{(t)}}\right)^{2p} \leq \frac{\left\|r^{(T)}\right\|_q^q}{\left\|r^{(0)}\right\|_q^q}\max_i\left(\sum_{t\in T_{lo},\alpha_i^{(t)}>1}\sqrt{\alpha_i^{(t)}}\right)^{2p}$$

$$\leq \left(\frac{2q}{2q-1}\right)^q\max_i\left(\sum_{t\in T_{lo},\alpha_i^{(t)}>1}\sqrt{\alpha_i^{(t)}}\right)^{2p} \quad \text{(since } \left\|r^{(0)}\right\|_q = \frac{2q}{2q-1}\text{)}$$

$$\leq 2\max_i\left(\sum_{t\in T_{lo},\alpha_i^{(t)}>1}\sqrt{\alpha_i^{(t)}}\right)^{2p}, \quad \text{(since } q \geq 1\text{)}$$

$\square$

Therefore there exists $i$ such that

$$\left(\sum_{t\in T_{lo},\alpha_i^{(t)}>1}\sqrt{\alpha_i^{(t)}}\right)^{2p} \geq \frac{1}{2}\left(\frac{T_{lo}}{2}\right)^{2p},$$

which gives us

$$\sum_{t \in T_{lo}, \alpha_i^{(t)} > 1} \sqrt{\alpha_i^{(t)}} \geq \frac{T_{lo}}{2} \frac{1}{2^{\frac{1}{2p}}} \geq \frac{T_{lo}}{4}, \text{ since } p \geq 1.$$

This lemma enables us to upper bound $T_{lo}$.

**Lemma E.11.** *We have $T_{lo} \leq \Theta\left(\frac{S^{1/2}}{\ln S} \ln n + \ln n\right)$.*

*Proof.* From Lemma E.10 we know that there exists a coordinate $i$ for which

$$\sum_{t \in T_{lo}: \alpha_i^{(t)} > 1} \sqrt{\alpha_i^{(t)}} > \frac{T_{lo}}{4}. \tag{8}$$

Furthermore by definition for all iterates in $T_{lo}$ we have that pointwise $\alpha_i^{(t)} = \frac{r_i^{(t+1)}}{r_i^{(t)}} \leq S$ and $\alpha_i^{(t)} = \left(\gamma_i^{(t)}\right)^{1/q} \geq 2^{\frac{1}{q}}$. This enables us to lower bound the final value of $\left(r_i^{(T)}\right)^q$ which is a lower bound on $\left\|r^{(T)}\right\|_q^q$. More precisely, we have $\frac{r_i^{(t+1)}}{r_i^{(t)}} \geq \alpha_i^{(t)}$ thus

$$\left(r_i^{(T)}\right)^q \geq \left(r_i^{(0)}\right)^q \cdot \prod_{t \in T_{lo}: \alpha_i^{(t)} > 1} \left(\alpha_i^{(t)}\right)^q = \frac{2q-1}{2q} \cdot \frac{1}{n} \cdot \prod_{t \in T_{lo}: \alpha_i^{(t)} > 1} \left(\alpha_i^{(t)}\right)^q. \tag{9}$$

Now we can proceed to lower bound this $r_i$ i.e. we lower bound the product in (9) using the lower bound we have in (8).

Similarly to the previous section, the worst case behavior i.e. slowest possible increase in $\left(r_i^{(T)}\right)^q$ is achieved in one of the two extreme cases:

(i) the $\alpha_i^{(t)}$ are all minimized i.e. $\alpha_i^{(t)} = 2^{\frac{1}{q}}$ in which case $\Theta(\ln n)$ such terms are sufficient to make their product $\geq 2n \geq \frac{2qn}{2q-1}$, which means that we are done, since then we have $\left\|r^{(T)}\right\|_q^q \geq \left(r_i^{(T)}\right)^q \geq 1$; so setting $T_{lo} \geq \Theta(\ln n)$ is sufficient to make this happen;

(ii) all the entries are maximized, i.e. $\alpha_i^{(t)} = S$ in which case we have that their product to power $q$ is at least $S^{\frac{T_{lo}q}{4S^{1/2}}} \geq 2n \geq \frac{2qn}{2q-1}$, so if we set $\frac{T_{lo}q}{4S^{1/2}} \ln S \geq \ln 2n$, ie, $T_{lo} \geq \frac{8S^{1/2}\ln(n)}{q \ln S}$, we guarantee that the corresponding $r_i$ increases to a value larger than 2. The fact that these two cases capture the slowest possible increase is shown in Lemma F.1.

Therefore we can set

$$T_{lo} = O\left(\frac{S^{1/2}}{\ln S} \ln n + \ln n\right).$$

$\square$

Finally, by the choice $S = n^{\frac{2}{2q+1}}$, we obtain the runtime guarantee.

**Lemma E.12.** *Algorithm 4 terminates in $O\left(n^{\frac{1}{2q+1}}\right)$ iterations.*

*Proof of Lemma E.4.* The proof of Lemma E.1 immediately follows from Lemmas E.7, E.8 and E.12.

$\square$

### E.2 PROOF OF LEMMA E.2

*Proof of Lemma E.2.* We define the function $\text{res}_x$ as follows

$$\text{res}_x(\Delta) = \langle g, \Delta \rangle - \langle R, \Delta^2 \rangle - \|\Delta\|_p^p$$

where $g = |x|^{p-2} x$, $R = 2|x|^{p-2}$. We use the following property of this function from Adil et al. (2019a; 2024): For $\lambda = 16p$ and for all $\Delta$

$$\|x\|_p^p - \left\|x - \frac{\Delta}{p}\right\|_p^p \geq \operatorname{res}_x(\Delta); \tag{10}$$

$$\|x\|_p^p - \left\|x - \lambda\frac{\Delta}{p}\right\|_p^p \leq \lambda\operatorname{res}_x(\Delta). \tag{11}$$

We prove the claim by induction.

For $t = 0$, we have $M^{(0)} := \frac{\left\|x^{(0)}\right\|_p^p}{16p} \geq \frac{\left\|x^{(t)}\right\|_p^p - \|x^*\|_p^p}{16p}$.

Now assume that we have $\left\|x^{(t)}\right\|_p^p - \|x^*\|_p^p \leq 16pM^{(t)}$. We have two cases.

Case 1. ResidualSolver returns an infeasibility certificate or ResidualSolver returns a primal solution $\tilde{\Delta}$ such that $\left\langle R^{(t)}, \tilde{\Delta}^2 \right\rangle \geq 2M^{(t)}$. In both scenarios, using Lemma E.1 we have

$$\min_{\substack{A\Delta=0 \\ \left\langle g^{(t)}, \Delta \right\rangle = \frac{M^{(t)}}{2}}} \left\|\Delta^2\right\|_{\frac{p}{2}} + (M^{(t)})^{\frac{2-p}{p}} \left\langle R^{(t)}, \Delta^2 \right\rangle \geq 2(M^{(t)})^{\frac{2}{p}}.$$

Hence for all $\Delta$ such that $A\Delta = 0$, $\left\langle g^{(t)}, \Delta \right\rangle = \frac{M^{(t)}}{2}$, either $\left\|\Delta^2\right\|_{\frac{p}{2}} \geq (M^{(t)})^{\frac{2}{p}} \Leftrightarrow \|\Delta\|_p^p \geq M^{(t)}$ or $(M^{(t)})^{\frac{2-p}{p}} \left\langle R^{(t)}, \Delta^2 \right\rangle \geq (M^{(t)})^{\frac{2}{p}} \Leftrightarrow \left\langle R^{(t)}, \Delta^2 \right\rangle \geq M^{(t)}$. For all $\Delta$ such that $A\Delta = 0$, we can write $\left\langle g^{(t)}, \Delta \right\rangle = a\frac{M^{(t)}}{2}$, for some constant $a \in \mathbb{R}$. We obtain either $\|\Delta\|_p^p \geq a^p M^{(t)}$ or $\left\langle R^{(t)}, \Delta^2 \right\rangle \geq a^2 M^{(t)}$, and thus for all $\Delta$

$$\operatorname{res}_{x^{(t)}}(\Delta) \leq M^{(t)}\left(\frac{1}{2}a - \min\left\{a^2, a^p\right\}\right) \leq \frac{M^{(t)}}{2} = M^{(t+1)}.$$

We write $\overline{\Delta} = \frac{x^{(t)} - x^*}{\lambda/p}$, for $\lambda = 16p$. Using property (11) of the $\operatorname{res}_x$, we have

$$\begin{aligned}
\left\|x^{(t+1)}\right\|_p^p - \|x^*\|_p^p &= \left\|x^{(t)}\right\|_p^p - \|x^*\|_p^p \\
&= \left\|x^{(t)}\right\|_p^p - \left\|x^{(t)} - \lambda\frac{\overline{\Delta}}{p}\right\|_p \\
&\leq \lambda\operatorname{res}_{x^{(t)}}\left(\overline{\Delta}\right) \\
&\leq 16pM^{(t+1)}.
\end{aligned}$$

Case 2. We have $\left\langle R, \tilde{\Delta}^2 \right\rangle < 2M^{(t)}$ and $\left\|\tilde{\Delta}\right\|_p \leq 4\sqrt{\kappa}(M^{(t)})^{\frac{1}{p}}$ and $\left\langle g, \tilde{\Delta} \right\rangle = \frac{M^{(t)}}{2}$

$$\begin{aligned}
\left\|x^{(t)}\right\|_p^p - \left\|x^{(t+1)}\right\|_p^p &= \left\|x^{(t)}\right\|_p^p - \left\|x^{(t)} - \frac{\tilde{\Delta}}{64p\kappa}\right\|_p^p \\
&\geq \operatorname{res}_{x^{(t)}}\left(\frac{\tilde{\Delta}}{64\kappa}\right) \\
&= \left\langle g, \frac{\tilde{\Delta}}{64\kappa} \right\rangle - \left\langle R, \left(\frac{\tilde{\Delta}}{64\kappa}\right)^2 \right\rangle - \left\|\frac{\tilde{\Delta}}{64\kappa}\right\|_p^p \\
&\geq \frac{M^{(t)}}{2^7\kappa} - \frac{M^{(t)}}{2^{11}\kappa^2} - \frac{M^{(t)}}{2^{4p}\kappa^{\frac{p}{2}}} \\
&\geq \frac{M^{(t)}}{2^7\kappa} - \frac{M^{(t)}}{2^{11}\kappa} - \frac{M^{(t)}}{2^8\kappa}, \qquad \text{(since } p \geq 2, \kappa \geq 1\text{)}
\end{aligned}$$

$$\geq \frac{M^{(t)}}{2^9 \kappa} \geq \frac{1}{2^{13} p \kappa} \left( \left\| x^{(t)} \right\|_p^p - \| x^* \|_p^p \right),$$

from which we obtain

$$\left\| x^{(t+1)} \right\|_p^p - \| x^* \|_p^p \leq \left\| x^{(t)} \right\|_p^p - \| x^* \|_p^p - \frac{1}{2^{13} p \kappa} \left( \left\| x^{(t)} \right\|_p^p - \| x^* \|_p^p \right)$$

$$\leq \left( 1 - \frac{1}{2^{13} p \kappa} \right) \left( \left\| x^{(t)} \right\|_p^p - \| x^* \|_p^p \right)$$

as needed. $\qquad \square$

## F LOWER BOUND LEMMA

**Lemma F.1.** *Let a set of nonnegative reals $\beta_1, \ldots, \beta_k$ such that $1 + \epsilon \leq \beta_i \leq S$, and $\sum_{i=1}^k \beta_i^{\frac{1}{r}} \geq K$, where $r \geq 2$. Then for any $k$ one has that*

$$\prod_{i=1}^k \beta_i \geq \min \left\{ S^{\frac{K}{S^{1/r}}}, (1 + \epsilon)^{\frac{K}{(1+\epsilon)^{1/r}}} \right\}.$$

*Proof.* Consider a fixed $k$, and let us attempt to minimize the product of $\beta_i$'s subject to the constraints. W.l.o.g. we have $\sum_{i=1}^k \beta_i^{\frac{1}{r}} = K$. Equivalently we want to minimize $\sum_{i=1}^k \log(\beta_i)$, which is a concave function. Therefore its minimizer is attained on the boundary of the feasible domain. This means that for some $0 \leq k' \leq k-1$, there are $k'$ elements equal to $1+\epsilon$, $k-1-k'$ equal to $S$, and one which is exactly equal to the remaining budget, i.e. $\left( K - k'(1+\epsilon)^{1/r} - (k-1-k')S^{1/r} \right)$, which yields the product $(1+\epsilon)^{k'} S^{k-k'-1} \cdot \left( K - k'(1+\epsilon)^{1/r} - (k-1-k')S^{1/r} \right)$. This can be relaxed by allowing $k$ and $k'$ to be non-integral. Hence we aim to minimize the product $(1+\epsilon)^{k'} S^{k-k'-1}$ subject to $k'(1+\epsilon)^{1/r} - (k-1-k')S^{1/r} = K$.

Finally, we observe that we can always obtain a better solution by placing all the available mass on a single one of the factors, i.e. we lower bound either by $S^{\frac{K}{S^{1/r}}}$ or $(1+\epsilon)^{\frac{K}{(1+\epsilon)^{1/r}}}$, whichever is lowest.

## G ITERATIVE REFINEMENT

In this section we provide a general technique for solving optimization problems to high-precision, by reducing then to an adaptive sequence of easier optimization problems, which only require approximate solutions. This formalizes the minimal requirements for the iterative refinement scheme employed in Adil et al. (2019a;b) to go through. We state the main lemma below.

**Lemma G.1.** *Let $\mathcal{D} \subseteq \mathbb{R}^n$ be a convex set, and let $f : \mathcal{D} \to \mathbb{R}$ be a convex function. Let $\eta \geq 0$ be a scalar, and suppose that for any $x \in \mathcal{D}$ there exists a function $h_x$ that approximates the Bregman divergence at $x$ in the sense that*

$$\frac{1}{\eta} h_x (\eta \delta) \leq f (x + \delta) - f (x) - \langle \nabla f (x), \delta \rangle \leq h_x (\delta) .$$

*Given access to an oracle that for any direction $v$ can provide $\kappa$-approximate minimizers to $\langle v, \delta \rangle + h_x (\delta)$ in the sense that it returns $\delta^\sharp$ such that $v + \delta^\sharp \in \mathcal{D}$ and*

$$\langle v, \delta^\sharp \rangle + h_x (\delta^\sharp) \leq \frac{1}{\kappa} \left( \min_{v + \delta \in \mathcal{D}} \langle v, \delta \rangle + h_x (\delta) \right),$$

*along with an initial point $x_0 \in \mathcal{D}$, in $O \left( \frac{\kappa}{\eta} \ln \frac{f(x_0) - f(x^*)}{\varepsilon} \right)$ calls to the oracle one can obtain a point $x$ such that $f (x) \leq f (x^*) + \varepsilon$, where $x^* \in \arg \min_{x \in \mathcal{D}} f (x)$.*

*Proof.* Let $\delta^\sharp$ be the a $\kappa$-approximate minimizer of $\langle \nabla f (x), \delta^\sharp \rangle + h_x (\delta^\sharp)$, which by definition satisfies:

$$\langle \nabla f (x), \delta^\sharp \rangle + h_x (\delta^\sharp) \leq \frac{1}{\kappa} \left( \min_{v + \delta \in \mathcal{D}} \langle \nabla f (x), \delta \rangle + h_x (\delta) \right). \tag{12}$$

Updating our iterate to $x' = x + \delta^\sharp$ we can bound the new function value as

$$f\left(x + \delta^\sharp\right)$$
$$= f\left(x\right) + \left\langle \nabla f\left(x\right), \delta^\sharp \right\rangle + h_x\left(\delta^\sharp\right) \qquad \text{(Bregman divergence upper bound)}$$
$$\leq f\left(x\right) + \frac{\eta}{\kappa}\left(\left\langle \nabla f\left(x\right), x^* - x \right\rangle + \frac{1}{\eta} h_x\left(\eta\left(x^* - x\right)\right)\right) \qquad \text{(using (12))}$$
$$= f\left(x\right) + \frac{\eta}{\kappa}\left(\left\langle \nabla f\left(x\right), x^* - x \right\rangle + \left(f\left(x^*\right) - f\left(x\right) - \left\langle \nabla f\left(x\right), x - x^* \right\rangle\right)\right)$$
$$\qquad \text{(Bregman divergence lower bound)}$$
$$= f\left(x\right) + \frac{\eta}{\kappa}\left(f\left(x^*\right) - f\left(x\right)\right),$$

from where we equivalently obtain that

$$f\left(x + \delta^\sharp\right) - f\left(x^*\right) \leq \left(1 - \frac{\eta}{\kappa}\right)\left(f\left(x\right) - f\left(x^*\right)\right).$$

Therefore to reduce the initial error $f\left(x_0\right) - f\left(x^*\right)$ to $\varepsilon$ it suffices to iterate $O\left(\frac{\kappa}{\eta} \ln \frac{f(x_0) - f(x^*)}{\varepsilon}\right)$ times. $\qquad \square$

The following lemma provides a sandwiching inequality for the Bregman divergence of $\|x\|_p^p$.

**Lemma G.2** (Adil et al. (2019b), Lemma B.1). *For any $x, \delta$ and $p \geq 2$, we have for $r = x^{p-2}$ and $g = px^{p-1}$,*

$$\frac{p}{8}\left\langle r, \delta^2 \right\rangle + \frac{1}{2^{p+1}}\|\delta\|_p^p \leq \|x + \delta\|_p^p - \|x\|_p^p - \langle g, \delta \rangle \leq 2p^2 \left\langle r, \delta^2 \right\rangle + p^p \|\delta\|_p^p.$$

As a corollary we see that the function $h_x\left(\delta\right) = 2p^2 \left\langle x^{p-2}, \delta^2 \right\rangle + p^p \|\delta\|_p^p$ satisfies the inequality required by Lemma G.1 for $\eta = \frac{1}{4p}$. We can thus conclude that given access to an oracle that approximately minimizes mixed $\ell_2 + \ell_p$ regression objectives, one can efficiently generate a high precision solution.

**Corollary G.1.** *Consider the $\ell_p$ regression problem $\min_{f:B^\top f=d} \|f\|_p^p$. Given access to an oracle that can compute $\kappa$-approximate minimizers to the optimization problem*

$$V^* := \min_{f:B^\top \Delta f=0} \left\langle pf^{p-1}, \Delta f \right\rangle + 2p^2 \left\langle f^{p-2}, \Delta f^2 \right\rangle + p^p \|\Delta f\|_p^p$$

*in the sense that it returns $\Delta f$ satisfying $B^\top \Delta f = 0$ and*

$$\left\langle pf^{p-1}, \Delta f \right\rangle + 2p^2 \left\langle f^{p-2}, \Delta f^2 \right\rangle + p^p \|\Delta f\|_p^p \leq \frac{1}{\kappa} V^*,$$

*along with an initial point $f_0$, satisfying $B^\top f = d$, in $O\left(\kappa p \ln \frac{\|f_0\|_p^p - \|f^*\|_p^p}{\varepsilon}\right)$ calls to the oracle one can obtain a point $f$ such that $\|f\|_p^p \leq \|f^*\|_p^p + \varepsilon$, where $f^* \in \arg\min_{B^\top f=d} \|f\|_p^p$.*

*Proof.* Using Lemma G.2 we verify that the function $h_f\left(\Delta f\right) = 2p^2 \left\langle f^{p-2}, \Delta f^2 \right\rangle + p^p \|\Delta f\|_p^p$ satisfies

$$\frac{1}{\eta} h_f\left(\eta \Delta f\right) \leq \|f + \Delta f\|_p^p - \|f\|_p^p + \left\langle pf^{p-1}, \Delta f \right\rangle \leq h_f\left(\Delta f\right)$$

for $\eta = \frac{1}{4p}$. Therefore by Lemma G.1 we can need $O\left(\kappa p \ln \frac{\|f_0\|_p^p - \|f^*\|_p^p}{\varepsilon}\right)$ iterations to obtain an $\varepsilon$-additive error to the regression problem. $\qquad \square$

$\qquad \square$

# H ADDITIONAL EXPERIMENTAL RESULTS

**Correctness of solution.** In Figure 3, we plot the error of the solutions outputted by our algorithm and $p$-IRLS against CVX in the random matrices and random graphs instances for $\epsilon = 10^{-10}$. In all cases, the error is below $\epsilon$.

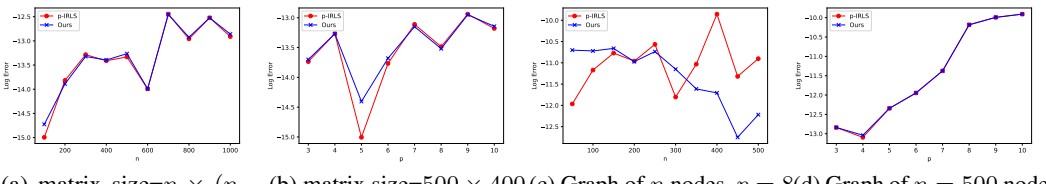

(a) matrix size=$n \times (n -$ (b) matrix size=$500 \times 400$ (c) Graph of $n$ nodes, $p = 8$ (d) Graph of $n = 500$ nodes
$50), p = 8$

Figure 3: Error of the solution against CVX/SDPT3 solution in log10 scale.

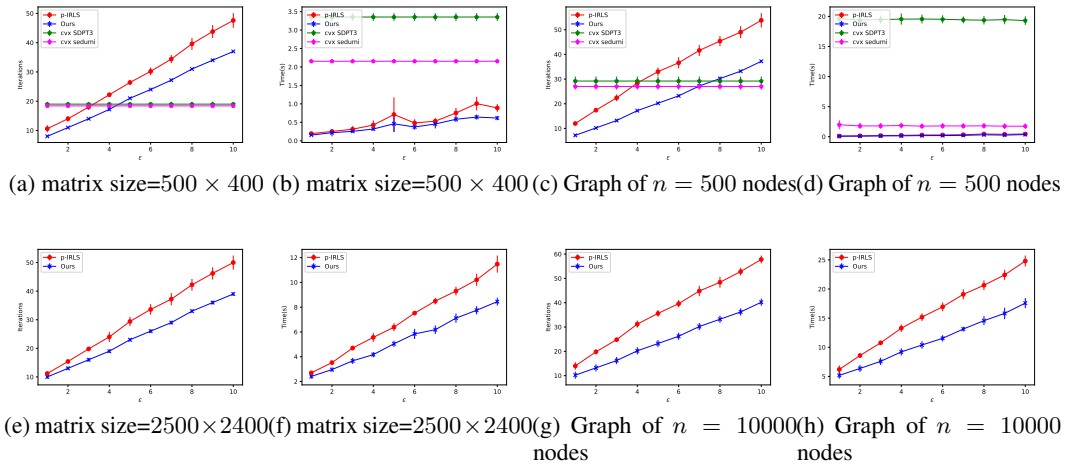

(a) matrix size=$500 \times 400$ (b) matrix size=$500 \times 400$ (c) Graph of $n = 500$ nodes (d) Graph of $n = 500$ nodes

(e) matrix size=$2500 \times 2400$ (f) matrix size=$2500 \times 2400$ (g) Graph of $n = 10000$ (h) Graph of $n = 10000$
nodes nodes

Figure 4: Performance when varying $\epsilon$ on random matrices and random graphs instances.

**When varying $\epsilon$.** In Figure 4, we plot iteration complexity and runtime in seconds of our algorithm, $p$-IRLS and CVX when varying $\epsilon$. Note that, CVX does not allow varying this parameter. In all experiment, we fix $p = 8$. For large instances, we only consider our solution against $p$-IRLS.

**For $1 < p < 2$.** In Figure 5, we plot iteration complexity and runtime in seconds of our algorithm, $p$-IRLS and CVX on random matrices of size $n \times (n - 100)$. We fix $\epsilon = 10^{-10}$. We test with $p = 1.1$ and $p = 1.9$.

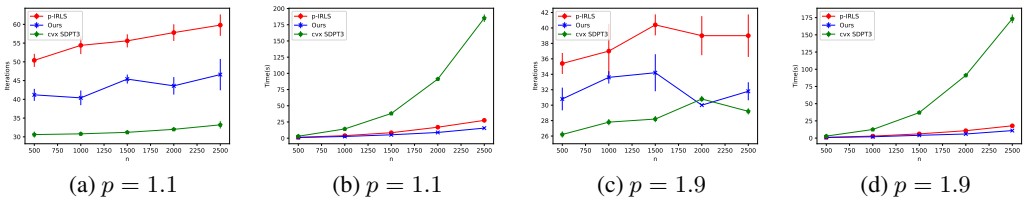

(a) $p = 1.1$  (b) $p = 1.1$  (c) $p = 1.9$  (d) $p = 1.9$

Figure 5: Performance when $p = 1.1$ and $p = 1.9$ on random matrices of size $n \times (n - 100)$.

