# OpenReview forum: "Improved $\ell_{p}$ Regression via Iteratively Reweighted Least Squares"
_ICLR.cc/2026/Conference — ICLR 2026 Poster_

### Official Review · Reviewer_BjD5 · 2025-10-29

**Soundness:** 3
**Presentation:** 3
**Contribution:** 3
**Rating:** 6
**Confidence:** 5

**Summary:**

The paper gives an algorithm for the p-norm regression problem for $p> 2$. The known algorithms include the algorithm by AKPS’ JACM 2024 which gives the best theoretical guarantees but a complicated algorithm, and a practical algorithm by APS’19 which outperforms all existing implementations at the cost of a worse iteration complexity. This paper bridges the gap by giving an algorithm that matches the guarantees of AKPS as well as outperforms the algorithm by APS’19.

Their algorithms is an IRLS style algorithm similar to the approach of APS’19 but their algorithm and analysis are quite different. The analysis follows a more primal dual approach similar to the works of Ene-Vladu ICML’2019. They first use an approach similar to EV’19 to give a low accuracy solver and then use the iterative refinement framework from AKPS to boost the accuracy. In order to do so they require modifying their algorithms to work for a regularized version of the problem.

**Strengths:**

1. Interesting theoretical result which can be of independent interest.
2. Bridges the gap between theory and practice for this problem.
3. They also test their algorithm on some real-world data sets which I dont think has been done before for this problem.

**Weaknesses:**

See questions. No major weaknesses!

**Questions:**

1. The algorithm looks more complicated than APS’19 and also of a similar level of complexity as AKPS JACM’24. It looks like a different approach, and maybe the authors could say something about the fundamental difference with previous works. I am not sold on the simplicity of the algorithm, regardless I think it is interesting as a standalone algorithm.
2. Also the difference in iteration counts in the experiments seems to be some fixed constant factor of maybe 2 or something. I am wondering if this is because APS’19 is not implemented well enough or this algorithm is genuinely faster due to the improved rates of convergence?

---

> ### Author Response · Authors · 2025-11-21
>
> We thank the reviewer for their hard work and valuable feedback.
>
> To answer to reviewer's questions.
>
> - Compared with AKPS JACM '24: We thank the reviewer for pointing out that our algorithm is fundamentally different from AKPS JACM '24. Both algorithms build on the residual problem solver but use completely different approaches for deriving the updates. AKPS JACM '24 use the Taylor expansion of the $\ell_{p}^{p}+\ell_{2}^{2}$ objective and derive their update based on it. In contrast, we use a primal-dual formulation of the problem and derive the update with the goal of maintaining invariant (1).
>
> - Regarding the simplicity of the algorithm. The core update step in Algorithm 4 is the computation of $\gamma$  (line 338), and the remaining steps are mainly for checking the conditions. The computation of $\gamma$  is done per-coordinate, which is simple in our opinion. Compared with APS '19, our algorithm maintains the simplicity but is quite significantly faster.
>
> - Regarding the implementability of the algorithms: The algorithm of AKPS uses theoretical choices for several parameters to guarantee the convergence rate. In practice, these parameters need to be tuned, which makes it difficult to deploy this algorithm in practice. Building on AKPS, APS '19 developed a theoretically slower yet more practical solver. The advantage of our algorithm is that it achieves the best of both worlds: it has convergence guarantee that matches AKPS, and it is easily implementable and achieves better empirical results than APS.
>
> - Regarding the performance comparison with APS '19. We use the provided implementation of their algorithm, so we believe the code is already optimized. We also use the same experimental setup (excluding the new experiments on the real-world datasets). To verify the benefit of our improved convergence rate, we show both the iteration counts and wall-clock time of the algorithms in our plots. For example in Figure 1(a) and 1(e), the number of iterations and wall clock time of our algorithm are both ~1.3x faster than those of p-IRLS. This improvement indeed comes from the improved convergence rate of our algorithm.

---

### Official Review · Reviewer_eXNj · 2025-10-29

**Soundness:** 4
**Presentation:** 4
**Contribution:** 4
**Rating:** 8
**Confidence:** 2

**Summary:**

In this paper, the authors present an algorithm for solving $\ell_p$ regression problems using the IRLS method. The algorithm is based on a clever update rule motivated by the dual formulation of an equivalent problem, and it achieves better complexity than other versions. In particular, the number of calls to the linear system solver matches the state-of-the-art theoretical order.

**Strengths:**

The paper is very well written in general. Although my expertise lies far from this area, I was still able to understand the main ideas and appreciate the contributions. The experimental results are very good.

**Weaknesses:**

This is not a weakness of the paper itself, but rather of this type of work in general. I believe that one of the strongest contributions of the paper lies in Theorems 1.1 and 1.2, but the proofs are relegated to the appendix (and I did not check them). I have always found that such papers are more appropriate for journal publication, where these key aspects can be better appreciated.

**Questions:**

I don't understand why to include the number of iterations of cvx sedumi/SDPT3 in Fig. 2. The nature of the iterations is quite different. I think it would be better to include them only in the time comparison.

---

> ### Author Response · Authors · 2025-11-21
>
> We thank the reviewer for their hard work and valuable feedback. Regarding the reviewer's question on the number of iterations of cvx sedumi/SDPT3, it is true that the nature of the iterations of cvx are different from those of p-IRLS and our algorithm. We can exclude the cvx algorithms from the iteration plots in the revision of the paper. The reviewer can refer to the wall-clock time plots for a fairer comparison.

---

### Official Review · Reviewer_3oUi · 2025-10-31

**Soundness:** 3
**Presentation:** 3
**Contribution:** 3
**Rating:** 6
**Confidence:** 2

**Summary:**

This paper presents new algorithms for solving ℓₚ regression problems using the Iteratively Reweighted Least Squares (IRLS) framework. The proposed methods achieve state-of-the-art iteration complexity while maintaining practical simplicity.

**Strengths:**

The paper establishes a solid theoretical foundation by matching the best-known asymptotic complexity of Adil et al. (JACM 2024) with a simpler iterative structure. Its primal-dual, invariant-based design offers a clean and principled acceleration of IRLS, effectively bridging theory and practice through provable guarantees and strong empirical performance. Experiments on synthetic and real datasets confirm notable gains in speed and accuracy, supported by clear and well-organized mathematical exposition.

**Weaknesses:**

None

**Questions:**

The algorithm’s performance under modern hardware acceleration is not discussed. I'm more interested in the parallel algorithm.

---

> ### Author Response · Authors · 2025-11-21
>
> We thank the reviewer for their hard work and valuable feedback.
>
> Regarding the question about parallelization and performance on modern hardware: Each iteration of our method is one linear system solve plus lower order operations that can be easily parallelized. Thus, the parallel depth of each iteration of our algorithm is determined by the depth of the underlying linear system solver. Additionally, the number of linear system solves required by our algorithm is state of the art and sublinear in $n$.
>
> Our algorithm is meant to leverage the existing linear system solvers. High quality parallel solvers exist for many structured linear systems, for example the Peng-Spielman parallel Laplacian solver (STOC 2014). Practical methods such as preconditioned conjugate gradient are also known to run efficiently on GPU clusters, as demonstrated by NVIDIA's large scale benchmarks (https://developer.nvidia.com/blog/optimizing-high-performance-conjugate-gradient-benchmark-gpus/). Moreover, any advances in parallel or hardware accelerated linear system solvers immediately yield low depth, hardware friendly algorithms for regression in our framework.

---

### Official Review · Reviewer_rkw2 · 2025-11-01

**Soundness:** 3
**Presentation:** 3
**Contribution:** 4
**Rating:** 6
**Confidence:** 3

**Summary:**

The paper studies the classic ℓp regression problem, which generalizes least squares (p = 2) and appears in robust regression, clustering, and semi-supervised learning. The authors propose a new Iteratively Reweighted Least Squares (IRLS)–based algorithm for ℓp regression that: Achieves the same theoretical iteration bound as the complex SODA 2019 / JACM 2024 algorithm of Adil–Kyng–Peng–Sachdeva,
But retains the simplicity and practical efficiency of the p-IRLS (NeurIPS 2019) method.

**Strengths:**

The IRLS method is widely used in practice but typically lacks strong theoretical guarantees. This paper proposes a variant that offers both rigorous theoretical foundations and favorable computational efficiency. The authors reformulate IRLS within a primal–dual invariant framework, deriving the update rule from a monotonic invariant of the dual objective. The algorithm adaptively rescales weights to preserve this invariant, achieving provably fast convergence for all values of p. The proposed approach is novel and potentially impactful beyond this specific setting, given the broad applicability of IRLS across various domains.

**Weaknesses:**

There should be more discussion on the intuition behind why this reformulation leads to improved convergence, as the underlying mechanism is not immediately clear—especially in contrast to the standard IRLS method, which is more intuitive.

The case of p<1 is particularly important due to its relevance in applications such as robust regression. However, this scenario is not addressed in the main paper and is only briefly mentioned in Appendix C, where the conclusions remain unclear. Given its significance, this case should be discussed in the main text, with the key findings and implications explicitly stated.

**Questions:**

The authors should include a comparison with existing IRLS algorithms and their variants, along with an explanation of why the proposed approach outperforms these methods.

The paper presents two versions of the algorithm—one for the low-accuracy regime and another for the high-accuracy regime. A discussion on their practical usage would be valuable. In particular, it would be helpful to clarify under what conditions each version is preferable, and whether a hybrid strategy (e.g., using the low-accuracy regime for initialization followed by the high-accuracy regime for refinement) would offer additional benefits.

---

> ### Author Response · Authors · 2025-11-21
>
> We thank the reviewer for their hard work and valuable feedback. We address the reviewer's questions below.
>
> 1. why this reformulation leads to improved convergence compared with existing IRLS algorithms
>
> As we describe in more detail in Sections 2.2 and 3.2, our primal-dual formulation of the problem allows us to derive an update from first principles based on maintaining the invariant (1) and the two bounds for the increase in the energy (Eq. (2)) and $\left\Vert r\right\Vert _{q}$ (Eq. (3)). As long as the algorithm maintains the invariant (1) with each update, it will reach a nearly-optimal dual objective value. Thus, to achieve a fast convergence, our algorithm also aims to ensure that $\left\Vert r\right\Vert _{q}$ increases very fast. To this end, we designed a novel update rule that increases $\left\Vert r\right\Vert _{q}$ as much as possible subject to maintaining the invariant. Our update rule allows us to understand how fast $\left\Vert r\right\Vert _{q}$ increases and eventually show our improved convergence guarantee.
>
> We would also like to note that, although the classical IRLS methods use a simple and intuitive update, they fail to converge in most settings such as for $p>3$. To the best of our knowledge, the p-IRLS algorithm of APS'19 is the first IRLS method that achieved a provably fast convergence for all $p\geq 2$. Our contribution is to show that it is possible to achieve even faster convergence than p-IRLS, and match the state of the art convergence achieved by non-IRLS methods.
>
> 2. The case of $1<p<2$
>
> Our algorithm can solve the problem to high precision in time $O\left(n^{\frac{q-2}{3q-2}}poly(\log n,\log\frac{1}{\epsilon},q)\right)=O\left(n^{\frac{2-p}{p+2}}poly(\log n,\log\frac{1}{\epsilon},\frac{p}{p-1})\right)$, where $q=\frac{p}{p-1}$ is the dual norm of $p$. This matches the state-of-the-art iteration complexity for the case $1<p<2$. We have added a theorem statement for this case to the appendix. For $p<1$ the problem becomes non-convex, and we do not consider this setting in our work.
>
> 3. Low vs high accuracy regimes
>
> The iteration complexity of the low accuracy algorithm is $O\left(n^{\frac{p-2}{3p-2}}poly\frac{1}{\epsilon}\right)$ while that of the high accuracy algorithm is $O\left(p^{2}n^{\frac{p-2}{3p-2}}poly\log\frac{1}{\epsilon}\right)$. There is a tradeoff between the dependence on $p$ and $\frac{1}{\epsilon}$. For the case with small $p$ and small $\epsilon$, the high accuracy algorithm is more favorable. On the other hand, for the case with large $p$ and large $\epsilon$, the low accuracy algorithm is more suitable.
>
> For our algorithm, initializing the high accuracy algorithm with a low accuracy solution can potentially improve the dependence of the iteration complexity on $p$ or $poly\log\frac{1}{\epsilon}$. However, we think the dependence on $n$ will not improve.

---

### Meta-Review · Area_Chair_Ge77 · 2025-12-10

**Summary:**

This paper proposes a new Iteratively Reweighted Least Squares (IRLS) algorithm for $\ell_p$ regression, achieving the same theoretical iteration complexity as the AKPS algorithm (JACM 2024) while remaining simple and efficient in practice. The algorithm is based on a primal-dual invariant framework that guides adaptive weight updates and leads to provably fast convergence across all values of $p$.

Reviewers found the paper clear, technically solid, and well motivated, with a clean theoretical formulation and strong empirical validation. They agreed that it successfully bridges the gap between the theory-heavy AKPS approach and the practical but less analyzable $p$-IRLS method. Some reviewers felt the intuition behind the new formulation could be explained more clearly, and noted that the important case of $p<1$ deserves more attention. Others requested discussion on practical use of the two algorithmic variants and on potential parallel implementations. Despite these points, the overall consensus is positive: the work is seen as a well-executed and meaningful contribution that combines theory and practicality effectively.

**Reviewer Concerns:**

I do not believe the question about $p<1$ has been resolved, but on the other hand, the setting is non-convex and likely requires different techniques.

**Reviewer Scores:**

It is not immediately clear to me that the reviewers would have changed their scores over the course of a full discussion phase.

---

### Decision · Program_Chairs · 2026-01-26

Accept (Poster)